# Catch-Only-One: Non-Transferable Examples for Model-Specific Authorization

## Abstract

Recent AI regulations call for data that remain useful for innovation while resistant to misuse, balancing utility with protection at the model level. Existing approaches either perturb data to make it unlearnable or retrain models to suppress transfer, but neither governs inference by unknown models, and both typically require control over training. We propose *non-transferable examples* (NEs), a training-free and data-agnostic input-side usage-control mechanism. We recode inputs within a model-specific low-sensitivity subspace, preserving outputs for the authorized model while reducing performance on unauthorized models through subspace misalignment. We establish formal bounds that guarantee utility for the authorized model and quantify deviation for unauthorized ones, with the Hoffman-Wielandt inequality linking degradation to spectral differences. Empirically, NEs retain performance on diverse vision backbones and state-of-the-art vision-language models under common preprocessing, whereas non-target models collapse even with reconstruction attempts. These results establish NEs as a practical means to preserve intended data utility while preventing unauthorized exploitation. Visual demos are available on our project page: https://github.com/model-specific/non-transferable-examples.

## 1 Introduction

*The title alludes to Joseph Heller's Catch-22, which symbolizes an unavoidable paradox. We adapt this notion in Catch-Only-One (or Catch-11) to capture the paradox that shared data may appear universally accessible yet remains usable only by a single authorized model.*

Recent regulatory initiatives—including the EU AI Act (European Parliament and Council, 2024), the US AI Action Plan (The White House, 2025), Australia's 2025 Privacy Law Reforms (Australian Productivity Commission, 2025), and Singapore's PDPC guidelines on AI and personal data (Tan, 2024)—emphasize that data should remain useful for licensed innovation while being shielded from misuse, whether in cyber attacks, CBRN (chemical, biological, radiological, and nuclear) applications, or unlicensed model training, echoing the Frontier Safety Framework recently proposed by DeepMind (Four Flynn, Helen King, Anca Dragan, 2025). In practice, however, this balance is far from realized. Once online, content is easily scraped, aggregated, and repurposed without consent: a few paintings can be used to clone an artist's style (Heikkil, 2022; Gal et al., 2022); medical scans shared for research can be exploited in membership inference attacks (Shokri et al., 2017); and billions of photos have been absorbed into training sets without license, fueling global disputes (The Hollywood Reporter, 2025). The stakes are tangible: a recent class action compelled Anthropic Inc. (Anthropic, 2025) to pay over \$1.5 billion and erase pirated training data (The Guardian, 2025). These cases underscore a pressing gap: while regulations demand that data retain authorized utility while blocking unauthorized use, no existing safeguard enforces this balance at the model level.

Three research directions have sought to mitigate unauthorized data use, ordered by increasing strength of protection. **Anti-learnability (unlearnability)** perturbs data before release so that standard training pipelines fail to converge (Ye & Wang, 2024; Wang et al., 2025a), preserving human perception but offering protection only against training misuse. **Ungeneralizable training** alters objectives or weights to suppress transfer in specific domains (Wang et al., 2022; Hong et al., 2025), but this requires retraining with custom losses and control of the pipeline, confining protection to modified models. **Fully homomorphic encryption (FHE)** (Gentry, 2009) guarantees maximal con-

fidentiality by enabling inference on encrypted data, but its extreme computational and memory costs make it impractical for routine applications such as online media, healthcare workflows, or MLaaS services (Ribeiro et al., 2015).

The three aforementioned approaches illustrate a spectrum of trade-offs: anti-learnability is lightweight but ineffective at inference, ungeneralizable training provides targeted suppression but relies on retraining, and FHE ensures the strongest confidentiality but remains prohibitively expensive in practice. These limitations highlight the need for methods that act directly at inference, preserving authorized utility while preventing post-release misuse, without imposing heavy retraining or cryptographic cost.

**Our Work**. We tackle the practical reality that, once data leaves its owner, it may be consumed by countless models. Our goal is to transform the data into a cipher that only a single authorized model $f^\star$ can interpret. We present a lightweight procedure that strategically recodes the data so it remains fully usable for $f^\star$ while withholding utility from any unauthorized models. This mechanism has broad applicability across online services such as machine learning as a service (MLaaS) (Ribeiro et al., 2015; Wang et al., 2025b), where uploaded data is often copied, cached, and reused beyond its original purpose (Gal et al., 2022; Liu et al., 2024a).

We propose *non-transferable examples* (NEs), a data-side construction that enforces *model-specific data usability* and requires neither retraining nor access to non-target models. NEs leverage a structural property of neural networks in which many input directions have negligible effect on early features, yielding a model-specific set of *insensitivity directions* that rarely align across models. With a small probe budget, we estimate the spectral basis for the $f^\star$'s insensitivity subspace of the authorized model, and add a calibrated perturbation confined to this basis, preserving the target's predictions while substantially reducing utility on unauthorized models.

We establish a formal foundation for NEs by deriving crucial bounds that provide strong theoretical guarantees: recoding within the identified insensitivity subspace preserves $f^\star$'s outputs within a quantified tolerance for authorized utility. For authorized-utility retention, we bound the distance between outputs computed with the same first-layer weight matrix $W$ via matrix perturbation theory, using random matrix and vector norm inequalities (Vershynin, 2018). For unauthorized usage, we prove a cross-model deviation bound using the Hoffman-Wielandt spectral inequality (Bhatia & Elsner, 1994), linking non-target degradation to differences in the singular spectrum and to misalignment between the corresponding feature subspaces.

We conduct a comprehensive empirical evaluation to demonstrate generality across tasks and model architectures. For image classification, we cover representative families including `ResNet`, `ViT`, `SwinV2`, `DeiT`, and `MambaVision`; recoded data remain usable only on the authorized target, while models with different weights collapse to effectively unusable performance (*e.g.,* even under an extreme distortion of $0\,\mathrm{dB}$ PSNR on ImageNet, the top-1 accuracy of the authorized ResNet-50 changes negligibly from 80.3% to 80.1%, whereas all other models drop to $\approx 0.1\%$). We also evaluate the vision-language models `Qwen2.5-VL` and `InternVL3` on the comprehensive benchmark MMBench (Liu et al., 2024b), which spans mathematical reasoning, chart/table understanding, document QA, and OCR, demonstrating applicability beyond classification to multimodal data. We further test NEs' resistance to common preprocessing and reconstruction attacks. Across all settings, NEs preserve authorized performance while rendering non-target systems effectively unusable.

**Contributions**. We summarize our main contributions as follows:

- **A new problem setting.** We formulate *model-specific data authorization*: user-provided data should remain fully usable for an authorized model while withholding utility from any other models. We introduce *non-transferable examples (NEs)*, a lightweight, training-free, data-agnostic spectral recoding method.

- **A formal framework and theoretical analysis.** We establish a strong formal foundation for NE, providing crucial bounds that guarantee efficacy and provide an independent, model-level confirmation of provable data-authorization guarantees.

- **An empirical evaluation.** Across diverse model architectures and data modalities, NEs consistently confine utility to the authorized model while non-targets fail, and they remain robust under routine preprocessing and resistant to reconstruction attempts, supporting practical deployment in real-world settings.

## 2 PROBLEM FORMULATION

We consider a supervised task with ambient input space $\mathcal{X}$ and output space $\mathcal{Y}$. A dataset $\mathcal{D}$ induces the task domains $\mathcal{X}_\mathcal{D} \subset \mathcal{X}$ and $\mathcal{Y}_\mathcal{D} \subset \mathcal{Y}$ observed in practice. Training on $\mathcal{D}$ yields a family of models $\mathcal{F}_\mathcal{D}$, where each $f \in \mathcal{F}_\mathcal{D}$ implements a mapping $f : \mathcal{X}_\mathcal{D} \to \mathcal{Y}_\mathcal{D}$. To quantify the *usability* of $f$ on inputs $x \in \mathcal{X}_\mathcal{D}$, we adopt a performance metric $m : \mathcal{F}_\mathcal{D} \times \mathcal{X} \to \mathbb{R}$ (*e.g.,* accuracy indicator for classification, negative sequence error for OCR, log-likelihood for generative decoding) and evaluate $\mathbb{E}_{x \in \mathcal{X}_\mathcal{D}}[m(f, x)]$. We postpone the detailed neural-network parameterization and notation (architectures, layers, operators) to Appendix B.1, where we formalize the conventions used in subsequent sections. For clarity, we assume smaller values of $m(\cdot)$ indicate better usability.

### 2.1 MODEL-SPECIFIC DATA REPRESENTATION

At inference on new data outside the training set, the data provider (defender) seeks to release content (*e.g.,* images, speech, documents) that is correctly processed by a designated target model $f^\star$ while remaining unusable for any non-target models $f' \in \mathcal{F}_\mathcal{D} \setminus \{f^\star\}$, *i.e.,*

$$\mathbb{E}_{x \in \mathcal{X}_\mathcal{D}}[m(f', x)] - \mathbb{E}_{x \in \mathcal{X}_\mathcal{D}}[m(f^\star, x)] \gg 0$$

for non-authorized aims.

Accordingly, we formulate a *data usage-control* objective: preserve the usability of $f^\star$ while degrading that of any non-target model, without requiring retraining and without imposing assumptions on non-target models, to maintain generality. Formally, we allow an input recoding $\tilde{x} = \mathcal{T}(x)$ with $\mathcal{T} : \mathcal{X} \to \mathcal{X}$. The resulting set $\widetilde{\mathcal{X}}_\mathcal{D} \subset \mathcal{X}$ is termed *non-transferable examples* (NEs).

We view NEs as a *model-specific data representation*: recoded inputs $\tilde{x} = \mathcal{T}(x)$ are tailored to the designated target $f^\star$ so that *authorized-utility retention* holds for $f^\star$ and *unauthorized-utility degradation* holds for any $f' \in \mathcal{F}_\mathcal{D} \setminus \{f^\star\}$. Formally, we state the following problem.

**Problem** (Model-specific Data Representation). *Given an authorized-utility tolerance $\rho \geq 0$ and a non-target separation margin $\gamma > 0$, a recoded data sample $\tilde{x} = \mathcal{T}(x)$ (for fixed $\mathcal{T}$) is* model-specific *to $f^\star$ if, for the designated $f^\star$ and any non-authorized $f'$, it satisfies:*

$$\begin{align} \textit{(authorized-utility retention)} \quad & \mathbb{E}_{\tilde{x} \in \widetilde{\mathcal{X}}_\mathcal{D}}[m(f^\star, \tilde{x})] - \mathbb{E}_{x \in \mathcal{X}_\mathcal{D}}[m(f^\star, x)] \leq \rho, \tag{1} \\ \textit{(unauthorized-utility degradation)} \quad & \mathbb{E}_{\tilde{x} \in \widetilde{\mathcal{X}}_\mathcal{D}}[m(f', \tilde{x})] - \mathbb{E}_{\tilde{x} \in \widetilde{\mathcal{X}}_\mathcal{D}}[m(f^\star, \tilde{x})] \geq \gamma. \tag{2} \end{align}$$

Before proceeding, we specify the threat model that this work takes into consideration.

### 2.2 THREAT MODEL

We consider a data producer (defender) who has white-box access to the authorized model $f^\star$ and a probe source from the task domain $\mathcal{X}_\mathcal{D}$ (a clean test set $\mathcal{D}$ or a sampler over $\mathcal{X}_\mathcal{D}$). The defender may query internal features of $f^\star$ and deploy an input-side transformation $\mathcal{T} : \mathcal{X} \to \mathcal{X}$, releasing only recoded inputs $\tilde{x} = \mathcal{T}(x)$. The algorithmic form of $\mathcal{T}$ may be public, while target-specific parameters (*e.g.,* spectral basis, thresholds) remain private. The defender has no access to non-target models in $\mathcal{F}_\mathcal{D} \setminus \{f^\star\}$.

Adversaries operate *unauthorized* models $f' \in \mathcal{F}_\mathcal{D} \setminus \{f^\star\}$ and receive only $\tilde{x}$. Before inference, they may optionally apply standard acquisition or preprocessing to obtain a preprocessed input $\tilde{x}'$ from $\tilde{x}$. The exact parameters of $\mathcal{T}$ are hidden. The adversary's objective is to *minimize* the task metric $m(f', \tilde{x}')$ under fixed compute or query budgets.

Specifically, we consider three adversary classes:

- (*i*) *General Adversary (GA):* Any non-target $f'$ with arbitrary architecture and parameters, given $\tilde{x}$ and allowed to apply an input-side preprocessing operator $\mathcal{A}$.
- (*ii*) *Transfer-match Adversary (TA):* A GA sharing the architecture of $f^\star$ but with different weights.
- (*iii*) *Adaptive Adversary (AA):* A GA/TA that optimizes over $\mathcal{A} \in \mathbb{A}$ to reduce $m(f', \mathcal{A}(\tilde{x}))$.

**Defense Objective**. Design $\mathcal{T}$ to achieve *model-specific data usability*, *i.e.,* to satisfy Formula (1) and Formula (2).

**Scope**. This setting is practical as an authorized third party may enforce such a process on behalf of the model owner: with agreed access to $f^\star$ (*e.g.*, feature probes sufficient to instantiate $\mathcal{T}$), it applies the protection server-side and releases only $\tilde{x}$. Neither the internals of $f^\star$ (architecture, weights) nor the target-specific parameters of $\mathcal{T}$ are disclosed to end users (see Appendix A for evidence).

## 3 OUR METHOD

Neural networks typically begin with a linear feature extractor (*e.g.*, a convolution with weight sharing or a token embedding), which reduces redundancy because input coordinates are often correlated (see Appendices B.2 and B.3). This motivates *input-side* perturbations that lie in an *insensitivity* subspace of the target model's first linear map: these directions are nearly inert for the target but, due to subspace misalignment across models, can induce nontrivial changes for non-targets.

### 3.1 INSENSITIVITY SUBSPACE IDENTIFICATION

Let $W$ denote the first linear transformation (bias omitted) of the target model. In practice, $W$ typically has a nontrivial nullspace $\mathrm{Null}(W) = \{x \in \mathcal{X} \mid Wx = 0\}$, owing to high input dimensionality (see Appendix B.4). For any $x \in \mathcal{X}$ and perturbation $\delta \in \mathrm{Null}(W)$, we have $W(x + \delta) = Wx$, so downstream computation receives identical features, *i.e.*, $f(x + \delta) = f(x)$.

To generalize the construction, we relax exact nulling and consider a low-sensitivity subspace that permits controlled feature deviation, *i.e.*, require $W\delta \approx 0$ rather than $W\delta = 0$. Perturbations confined to such an insensitivity subspace induce only small changes beyond the first layer, especially given intervening nonlinearities (*e.g.*, ReLU truncation, sigmoid/tanh saturation). To identify such directions for a given $W \in \mathbb{R}^{m \times n}$, we consider its singular value decomposition (SVD) $W = U\Sigma V^\top$: the nullspace is spanned by right singular vectors with zero singular values, and by extension we define the $\tau$-*insensitive subspace* as the span of right singular vectors whose singular values are at most a threshold $\tau > 0$. We formalize it as follows.

**Definition 1** ($\tau$-insensitive Subspace). *Let $W \in \mathbb{R}^{m \times n}$ have SVD, $W = USV^\top$, with singular values ordered $s_1 \leq s_2 \leq \cdots \leq s_n$. Given a spectral threshold $\tau > 0$, the $\tau$-insensitive subspace is*

$$\mathrm{Ins}_\tau(W) = \mathrm{span}\{(v_1, v_2, \ldots, v_k) \mid \sum_{i=1}^k s_i \leq \tau, \, W = USV^\top\}, \tag{3}$$

*where $v_i$ denotes the $i$-th column of $V$ corresponding to the $i$-th singular value $s_i$ in $S$.*

This construction naturally captures the nullspace, and $\mathrm{Ins}_\tau(W) \supseteq \mathrm{Null}(W)$ for any $\tau \geq 0$.

### 3.2 NON-TRANSFERABLE EXAMPLES

Inheriting Definition 1, and given a $\tau$-insensitive subspace $\mathrm{Ins}_\tau(W)$, we first sample a vector $z \in \mathbb{R}^n$ (*e.g.*, *i.i.d.* Gaussian, structured pattern, or content-dependent code) and zero out the coordinates aligned with singular values exceeding $\tau$ (*i.e.*, directions sensitive to the target model). Then, we project $z$ onto $\mathrm{Ins}_\tau(W)$ to obtain a perturbation $\delta$ in input space by $\delta = Vz$. Following that, the perturbation $\delta$ is added to the original input $x$ to obtain a recoded input $\tilde{x} = x + \delta$, which forms an NE. Note that $V$ is an orthogonal matrix such that $V^\top V = V^{-1}V = I$. Intuitively, we have $W\delta = WVz = USV^\top Vz = USz$, where $Sz$ has small entries since $z$ only has non-zero elements on coordinates with small singular values. By the distributive law of matrix multiplication, $W\tilde{x} = W(x + \delta) = Wx + W\delta$, where $W\delta$ is small and thus $W\tilde{x}$ is close to $Wx$.

**Practical Implementation**. Our procedure is model-agnostic and applies across architectures. Fully connected fronts can directly use their first weight matrix as $W$. For convolutional fronts, we apply the construction to the linearized operator of the first convolution (*e.g.*, `nn.unfold`) to obtain $W$; after synthesis, the perturbation is folded back to the native input layout. For Transformer-style models (*e.g.*, BERT (Devlin et al., 2019)), we take $W$ to be the input projection of the first multi-head self-attention block *after* the embedding layer (*e.g.*, the `QKV` input projection or its concatenated form) and apply the same spectral construction. In this work, we instantiate and evaluate the approach on convolutional and Transformer architectures. A theoretical discussion of the equivalence and generalizability of these instantiations is provided in Appendices B.5 and B.6. The generation

of the vector $z$ can be stochastic (*e.g.,* seeded per instance) or deterministic (*e.g.,* fixed codes per class or per client). We use $\delta \leftarrow \lambda \cdot \delta / \max\{1, \|\delta\|_2\}$ to ensure $\|\delta\|_2 \leq 1$ and then introduce a scaling factor $\lambda > 0$ to scale its amplitude. The parameters $\tau$ and $\lambda$ govern the trade-off between authorized-utility retention and unauthorized-utility degradation: larger $\tau$ tends to increase more perturbed input dimensions, while larger $\lambda$ increases the perturbation magnitude. In practice, the permissible insensitive space is deliberately loose, offering substantial flexibility in synthesis, as empirically explored in Section 5.

## 4 THEORETICAL ANALYSIS

This section provides a formal foundation for non-transferable examples (NEs) and establishes their *model-specific* nature via the two properties defined earlier: *authorized-utility retention* (Section 4.1) and *unauthorized-utility degradation* (Section 4.2). Consistent with classical analyses of perturbation propagation and local robustness (Huang et al., 2021; Wang et al., 2024; Qian & Klabjan, 2021; Ma et al., 2024b; 2025), we focus on the network's first linear transformation and treat subsequent layers implicitly.

### 4.1 AUTHORIZED-UTILITY RETENTION

We begin by quantifying how the recoded input $\tilde{x} = x + \delta$ affects the first-layer features of the *target* (authorized) model relative to the original input $x$ Throughout the analysis, $\| \cdot \|_F$ and $\| \cdot \|_2$ denote the Frobenius and Euclidean norms, respectively.

**Theorem 1** (Bounding Authorized Utility). *$W \in \mathbb{R}^{m \times n}$ represents a linear transformation as the first operation in a neural network, and its SVD is $W = USV^\top$. The perturbation $\delta = Vz$ in $\mathrm{Ins}_\tau(W)$ is generated following Section 3. $z \in \mathbb{R}^n$ is a random vector, each entry of which is i.i.d. by following a standard normal distribution $\mathcal{N}(0, \sigma)$. Let $\tilde{x} = x + \delta$ be the recoded input, the following bound holds,*

$$\|W\tilde{x} - Wx\|_2 < \tau \cdot \sqrt{k\sigma^2 + t}, \tag{4}$$

*with high probability of at least $1 - 2k\sigma^4/t^2$, where $t > 0$ is a small positive number.*

*Proof.* We first note that $\tilde{x} = x + \delta = x + Vz$. Consequently, $W\tilde{x} - Wx = W\delta = USV^\top Vz$. Since $V$ is orthogonal and $V^\top V$ is the identity, $USV^\top Vz = USz$.

Let $U = [u_1, u_2, \ldots, u_m]$ and assume each $s_q$ ($q \leq k$) satisfies $s_q \leq \tau$. Then

$$\|W\tilde{x} - Wx\|_F = \|US^*z\|_F = \sqrt{\sum_{p=1}^m (\sum_{q=1}^k u_{p,q} s_q z_q)^2} \leq \tau \sqrt{\sum_{p=1}^m (\sum_{q=1}^k u_{p,q} z_q)^2} = \tau \cdot \|Uz\|_F,$$

where $u_{p,q}$ is the $(p, q)$-th entry of the matrix $U$, $s_q$ is the $q$-th singular value in $S$, and $z_q$ is the $q$-th entry of $z$.

Using the orthogonality of $U$ (so that $U^\top U$ is the identity), we further obtain

$$\|Uz\|_F^2 = \mathrm{trace}((Uz)^\top (Uz)) = \mathrm{trace}(z^\top U^\top Uz) = \mathrm{trace}(z^\top z) = \|z\|_F^2 = \|z\|_2^2.$$

Hence, $\|W\tilde{x} - Wx\|_F \leq \tau \cdot \|z\|_2$.

Since $z$ is a random vector following the standard normal distribution $\mathcal{N}(0, \sigma)$, *i.e.,* each entry of $z$ is independently sampled from $\mathcal{N}(0, \sigma)$ so that $z_q \sim \mathcal{N}(0, \sigma)$, we have that $z_q^2$ follows the chi-squared distribution with one degree of freedom, *i.e.,* $z_q^2 \sim \chi^2(1)$, with $\mathbb{E}[z_q^2] = \sigma^2$ and $\mathrm{Var}(z_q^2) = 2\sigma^4$. Therefore, $\mathbb{E}[\|z\|_2^2] = \sum_{q=1}^k \mathbb{E}[z_q^2] = k\sigma^2$ and $\mathrm{Var}(\|z\|_2^2) = \sum_{q=1}^k \mathrm{Var}(z_q^2) = 2k\sigma^4$. By Chebyshev's inequality, we can bound the probability of the perturbation effect as

$$\mathbb{P}\{\|z\|_2^2 \geq k\sigma^2 + t\} \leq \mathrm{Var}(\|z\|_2^2)/t^2 = 2k\sigma^4/t^2,$$

where $t > 0$ is a small positive number. Combining this we conclude that such effect is bounded by $\tau \cdot \|z\|_2$ with high probability, *i.e.,* considering $\|W\tilde{x} - Wx\|_2 \leq \|W\tilde{x} - Wx\|_F$,

$$\mathbb{P}\{\|W\tilde{x} - Wx\|_2 < \tau \cdot \sqrt{k\sigma^2 + t}\} > 1 - 2k\sigma^4/t^2.$$

$\square$

It is worth highlighting that Theorem 1 ensures the perturbation effect on the first linear transformation can be kept small via appropriate choices of $\tau$ (which determines $k$) and $\sigma$. Given the generalization behavior of modern neural networks, such small first-layer perturbations are typically attenuated or tolerated by subsequent layers (Novak et al., 2018). Moreover, activation nonlinearities such as ReLU (zeroing negatives) or saturating sigmoids/tanh can further dampen perturbation propagation in practice, since inactive neurons contribute zero outputs for negative inputs. Together, these considerations evidently imply that $|f^\star(\tilde{x}) - f^\star(x)|$ remains small, thereby validating Formula (1) in Section 2.

## 4.2 Unauthorized-utility Degradation

We now turn to the cross-model effect and bound the discrepancy between the outputs of the *first* linear transformations in two models that differ in initialization or training. Intuitively, a perturbation constructed from the insensitivity subspace of one model can substantially impact another model, precisely because their first-layer SVDs differ. Formally, consider two models with first-layer weight matrices $W_1$ and $W_2$, as stated below; the theorem below is based on the classical Hoffman-Wielandt inequality (Bhatia & Elsner, 1994). For brevity, we defer detailed assumptions and supporting numerical experiments to Appendix C.

**Theorem 2** (Bounding Unauthorized Utility). *Given two models with the same-sized first-layer weight matrices $W_1, W_2 \in \mathbb{R}^{m \times n}$ trained on the same dataset, their SVDs are $W_1 = U_1 S_1 V_1^\top$ and $W_2 = U_2 S_2 V_2^\top$. Let $V_1 = [v_{1,1}, v_{1,2}, \ldots, v_{1,n}]$ and $V_2 = [v_{2,1}, v_{2,2}, \ldots, v_{2,n}]$ be the right singular vector matrices of $W_1$ and $W_2$. The perturbation is $\delta = Vz$, $\delta \in \mathrm{Ins}_\tau(W)$, generated following Section 3. $z \in \mathbb{R}^n$ is a random vector, each entry of which is i.i.d. by following a standard normal distribution $\mathcal{N}(0, \sigma)$. Let $\tilde{x} = x + \delta$ be the recoded input. Given any $\sigma_{1,i} \leq \tau$ $(1 \leq i \leq k)$ by Definition 1, the following bound holds,*

$$\|(\sigma_{1,i} v_{1,i} - \sigma_{2,i} v_{2,i})^\top \tilde{x}\|_2 \leq \|\tilde{x}\|_2 (\tau \|W_1 - W_2\|_2/\varepsilon + \varepsilon), \tag{5}$$

*where $\varepsilon = \min(|\sigma_{1,i} - \sigma_{2,j}| \mid \forall i, j = 1, 2, \ldots, n)$ is the minimum gap between the $i$-th singular value of $W_1$ and the singular values of $W_2$.*

*Proof.* We begin with the decomposition

$$\|(\sigma_{1,i} v_{1,i} - \sigma_{2,i} v_{2,i})^\top \tilde{x}\|_2$$
$$\leq \|(\sigma_{1,i} v_{1,i} - \sigma_{1,i} v_{2,i})^\top \tilde{x} + (\sigma_{1,i} v_{2,i} - \sigma_{2,i} v_{2,i})^\top \tilde{x}\|_2$$
$$= \|\sigma_{1,i}(v_{1,i} - v_{2,i})^\top \tilde{x}\|_2 + \|(\sigma_{1,i} - \sigma_{2,i})(v_{2,i})^\top \tilde{x}\|_2$$
$$\leq |\sigma_{1,i}| \|v_{1,i} - v_{2,i}\|_2 \|\tilde{x}\|_2 + |\sigma_{1,i} - \sigma_{2,i}| \|v_{2,i}\|_2 \|\tilde{x}\|_2$$

Considering $\|v_{2,i}\|_2 = 1$ (orthogonality of $V_2$) and $\sigma_{1,i} \leq \tau$ (Definition 1),

$$\|(\sigma_{1,i} v_{1,i} - \sigma_{2,i} v_{2,i})^\top \tilde{x}\|_2 \leq \|\tilde{x}\|_2 (\tau \|v_{1,i} - v_{2,i}\|_2 + |\sigma_{1,i} - \sigma_{2,i}|).$$

By the Hoffman-Wielandt theorem (Bhatia & Elsner, 1994), we have

$$\|v_{1,i} - v_{2,i}\|_2 \leq \|W_1 - W_2\|_2/\varepsilon,$$

where $\varepsilon = \min(|\sigma_{1,i} - \sigma_{2,j}| \mid \forall i, j = 1, 2, \ldots, n)$ is the minimum gap between the $i$-th singular value of $W_1$ and the adjacent singular values of $W_2$, $v_{1,i}$ and $v_{2,i}$ are the $i$-th right singular vectors of $W_1$ and $W_2$, and $s_{1,i}$ and $s_{2,j}$ are the $i$-th and $j$-th singular values of $W_1$ and $W_2$, accordingly. Thus, we have, considering $|\sigma_{1,i} - \sigma_{2,i}| \leq \varepsilon$ because the singular values are in descending order,

$$\|(\sigma_{1,i} v_{1,i} - \sigma_{2,i} v_{2,i})^\top \tilde{x}\|_2 \leq \|\tilde{x}\|_2 (\tau \|W_1 - W_2\|_2/\varepsilon + \varepsilon),$$

which concludes the proof. $\square$

Theorem 2 is strong in that it operates at the level of individual singular values. It shows that a perturbation taken from the insensitivity subspace of one model can significantly affect the first-layer output of another model. This remains true even when $\delta$ is near zero (by Assumption 1; see Appendix C.1 for numerical evidence and further details), so the effect is driven by structural differences rather than by large input distortion. Moreover, under Assumption 2 (see Appendix C.2), both models derive their first-layer weights from the same dataset and their right singular vectors

align with principal components of the data representations. Because spectral flatness permits multiple feasible principal-component solutions, there are correspondingly multiple feasible first-layer weight matrices. As a result, when singular values multiply their associated directions, the first model (with $W_1$) has a small $\sigma_{1,i}$ and thus a near-zero $\sigma_{1,i} v_{1,i}^\top \tilde{x}$, whereas the second (with $W_2$) has a different $\sigma_{2,i}$ and a non-zero $\sigma_{2,i} v_{2,i}^\top \tilde{x}$. Consequently, $|f'(\tilde{x}) - f^\star(\tilde{x})|$ is large, which directly yields Formula (2) in Section 2.

## 5 EXPERIMENTS

We empirically verify whether our non-*transferable examples* (NEs) preserve the authorized model's utility, retaining clean accuracy and intended task behavior, while rendering unauthorized models unable. The evaluation proceeds in three parts: *(i)* cross-model non-transferability covering both cross model architecture (*GA* in Section 2.2) transfer and same model architecture with different weights (*TA*); *(ii)* head-to-head comparisons with representative baselines under matched conditions; and *(iii)* real-world practicality, including generalization across modalities as well as robustness to common preprocessing pipelines and reconstruction attacks (*AA*).

**Experimental Setup**. We evaluate five widely used ImageNet-pretrained backbones that span classic to recent designs: ResNet-50 (He et al., 2016) (convolutional neural network), ViT-base-patch16-22 (Dosovitskiy et al., 2021) (vision transformer with patch-wise tokens), SwinV2-tiny-patch4-window8-256 (Liu et al., 2021) (hierarchical transformer with pyramid features common in real systems), DeiT-base-patch16-224 (Touvron et al., 2021) (data-efficient training of a ViT with distillation), and MambaVision-T-1K (Hatamizadeh & Kautz, 2025) (state-space sequence model adapted for vision). Experiments are conducted on CIFAR-10 and ImageNet-1K; for CIFAR-10, models are fine-tuned for 10 epochs from the ImageNet-1K checkpoints. "Baseline" entries in the tables denote clean accuracy under our evaluation pipeline.

We also evaluate leading-edge vision-language transformers Qwen2.5-VL-3B-Instruct (Bai et al., 2025) and InternVL3-1B (Zhu et al., 2025) on the comprehensive MMBench (Liu et al., 2024b), covering mathematical reasoning, chart and table understanding, document question answering, and OCR, to mirror real-world usage that includes cross-modality generalization and robustness to common preprocessing.

### 5.1 CROSS-MODEL NON-TRANSFERABILITY

**NE Construction**. To select a suitable perturbation strength that preserves the authorized model's accuracy while sharply reducing unauthorized accuracy, we randomly sample 512 images from ImageNet and generate NEs at multiple perturbation levels, recording top-1 accuracy as a function of strength. As a comparator, we use an unauthorized model given by ResNet-50 fine-tuned for 10 epochs on CIFAR-10, and measure how inputs recoded for different target models perform on this same unauthorized model. Results are shown in Figure 1. Across all settings, unauthorized accuracy collapses to an unusable level by 25 to 20dB PSNR (peak-signal-to-noise ratio (Hore & Ziou, 2010), lower PSNR indicates stronger perturbation and higher perceptual distortion), while the autho-

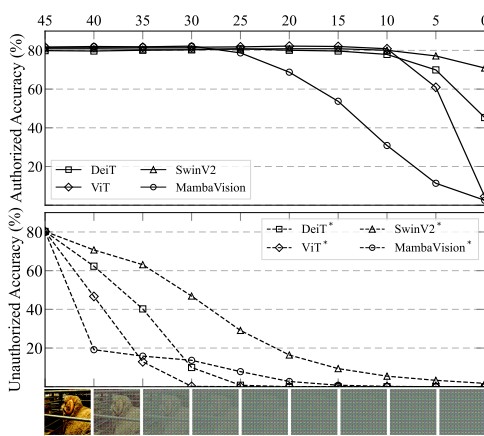

Figure 1: Authorized vs. unauthorized accuracy on target-recoded inputs across perturbation strength; see D.1 for visual clarification.

rized network experiences only a negligible drop. Several backbones remain stable even at $\leq$10dB, but we standardize on 20dB[1] for the rest of the experiments to use a conservative setting, as SwinV2-T shows a slight authorized drop below this point. We sample a recoding vector $z$ with the same dimension as the input representation with *i.i.d.* Gaussian entries, then project $z$ onto the $\tau$-insensitive

---

[1]We parameterize perturbation strength by PSNR for interpretability and fair comparison.

directions with $\tau = 10^{-4}$. For each authorized target $f^\star$, we report authorized accuracy on the recoded inputs $\tilde{x} = \mathcal{T}(x)$ and the accuracies of unauthorized models $f' \neq f^\star$ on the same $\tilde{x}$.

Table 1: Cross-model non-transferability on CIFAR-10 and ImageNet. Rows denote the target (authorized) model used to generate NEs; columns denote the evaluated model on the same recoded inputs. Green diagonal entries mark authorized accuracy; off-diagonals show unauthorized utility.

| | CIFAR10 | | | | | ImageNet | | | | |
|---|---|---|---|---|---|---|---|---|---|---|
| | ResNet-50 | ViT-B | SwinV2-T | DeiT-B | MambaVision-T | ResNet-50 | ViT-B | SwinV2-T | DeiT-B | MambaVision-T |
| Baseline | 98.2% | 98.8% | 96.1% | 96.1% | 96.9% | 80.3% | 81.6% | 80.9% | 79.9% | 82.4% |
| ResNet-50 | **97.7%** | 12.5% | 13.6% | 9.4% | 10.0% | **80.2%** | 0.0% | 0.1% | 0.1% | 0.1% |
| ViT-B | 10.5% | **98.7%** | 9.3% | 12.1% | 9.7% | 0.0% | **81.3%** | 0.0% | 0.0% | 0.0% |
| SwinV2-T | 15.6% | 11.7% | **88.4%** | 20.6% | 18.8% | 9.1% | 4.3% | **71.7%** | 7.4% | 12.9% |
| DeiT-B | 9.4% | 9.8% | 9.8% | **96.1%** | 5.5% | 0.0% | 0.0% | 0.0% | **79.3%** | 1.0% |
| MambaVision-T | 17.6% | 7.0% | 13.7% | 11.3% | **94.5%** | 5.8% | 0.0% | 1.5% | 0.7% | **81.0%** |

Table 1 reports a $5 \times 5$ cross-architecture matrix on CIFAR-10 and ImageNet. At 20dB PSNR, NEs keep authorized performance close to clean (*e.g.,* from 98.8% to 98.7% on CIFAR-10; from 80.3% to 80.2% on ImageNet), while unauthorized models collapse to chance-level utility (off-diagonals around 5.5–20.6% on CIFAR-10 and 0.0–12.9% on ImageNet). SwinV2 exhibits a slightly larger authorized drop on ImageNet (from 80.9% to 71.7%), which we attribute to sensitivity in its patch-merging pipeline; modest tuning of basis selection fixes this in practice. Crucially, in all cases, the off-diagonal entries remain at unusable accuracy, demonstrating strong architecture-specific non-transferability. Holding the architecture fixed but changing the weights, NE constructed with one weight set does not transfer to the same architecture with a different weight set. Along the diagonal of Table 2, unauthorized accuracy stays at a completely unusable level on both datasets for all backbones. Off-diagonal entries (cross-architecture) are shown in grey for completeness and mirror the behavior in Table 1. These results underscore strong model specificity: non-transferability holds across different architectures and across weight variants of the same architecture.

Table 2: Model-specific non-transferability. Diagonal entries compare the same model architecture with different weights; shaded off-diagonals are cross-architecture and included for completeness.

| | | CIFAR10 | | | | | | ImageNet | | | | |
|---|---|---|---|---|---|---|---|---|---|---|---|---|
| | | ResNet-50 | ViT-B | SwinV2-T | DeiT-B | MambaVision-T | | ResNet-50 | ViT-B | SwinV2-T | DeiT-B | MambaVision-T |
| ResNet-50 | ImageNet | 13.3% | 9.4% | 8.2% | 9.4% | 7.8% | CIFAR10 | 1.2% | 0.0% | 4.4% | 0.0% | 0.0% |
| ViT-B | | 10.1% | 9.6% | 10.1% | 11.5% | 9.8% | | 0.0% | 0.0% | 0.0% | 0.0% | 0.0% |
| SwinV2-T | | 10.5% | 10.2% | 21.0% | 11.3% | 14.5% | | 0.0% | 0.0% | 0.0% | 0.0% | 0.0% |
| DeiT-B | | 12.5% | 9.3% | 7.0% | 14.8% | 10.9% | | 0.0% | 0.0% | 0.0% | 0.0% | 0.0% |
| MambaVision-T | | 14.5% | 5.1% | 12.5% | 9.8% | 8.2% | | 0.0% | 0.0% | 1.9% | 0.0% | 0.0% |

### 5.2 Baseline Comparison

We compare the NE with authorization-oriented baselines that restrict model use by altering training, encrypting inference, or constraining the computation itself. Specifically, we include Differential Privacy (DP) (Dwork, 2006), which injects calibrated noise during training to limit extractable information; Fully Homomorphic Encryption (FHE), which executes inference over encrypted inputs and weights to protect access without changing model behavior; and AlgoSpec (Liu et al., 2024a), a specification-style approach that replaces nonlinear components with low-degree polynomial surrogates so the end-to-end pipeline conforms to a prescribed algorithmic specification intended to gate unauthorized use.

Table 3: Comparison with other authentication methods.

| | CIFAR-10 | | | | | ImageNet | | | | |
|---|---|---|---|---|---|---|---|---|---|---|
| | Plain | DP | FHE[3] | AlgoSpec | NE (Ours) | Plain | DP | FHE[3] | AlgoSpec | NE (Ours) |
| ResNet-50 | 98.2% | 59.8%[1] | 87.8%[3] | 6.4% | 97.7% | 80.3% | 63.1%[1] | – | 0.1% | 80.2% |
| ViT-B | 98.8% | –[1] | – | 10.0% | 98.7% | 81.6% | –[1] | – | 0.0% | 81.3% |
| Protection | ✗ | ✗[2] | ✓ | ✗[2] | ✓ | ✗ | ✗[2] | ✓ | ✗[2] | ✓ |

[1] DP struggles with batch norm and does not support multi-head attention in Transformers.
[2] The authorized model performance is significantly impacted.
[3] Due to too long execution time, we only provide data of accuracy that we can find in public papers.

Results for ResNet-50 and ViT-B on CIFAR-10 and ImageNet are shown in Table 3. For DP training, we follow the setting of Li et al. (2024) and implement it with IBM `DiffPrivLib` (Holohan et al., 2019). In our setup, authorized accuracy drops markedly, with more than 30% loss on CIFAR-10 and around 20% on ImageNet for ResNet-50, largely due to interactions with batch normalization,

and our DP pipeline does not support transformer variants, which limits applicability in this setting. For FHE, we adopt a CKKS encrypted-inference configuration following TenSEAL (Benaissa et al., 2021); However, running it at our model and dataset scale was computationally infeasible: processing a single image took more than 30 minutes. We therefore cite published ResNet-20 CIFAR-10 results (Meftah et al., 2021), which preserve clean authorized accuracy but incur heavy computational cost, highlighting the practicality gap of FHE at scale. For ALGOSPEC, polynomial approximation of modern deep networks accumulates approximation error with depth and width, which in our experiments drives authorized accuracy toward random guess on both datasets. In sharp contrast, NE is a lightweight input-side recoding tied to the target model that adds negligible inference overhead, and under matched conditions preserves authorized accuracy while driving mean unauthorized accuracy to chance on both datasets.

## 5.3 REAL-WORLD PRACTICALITY

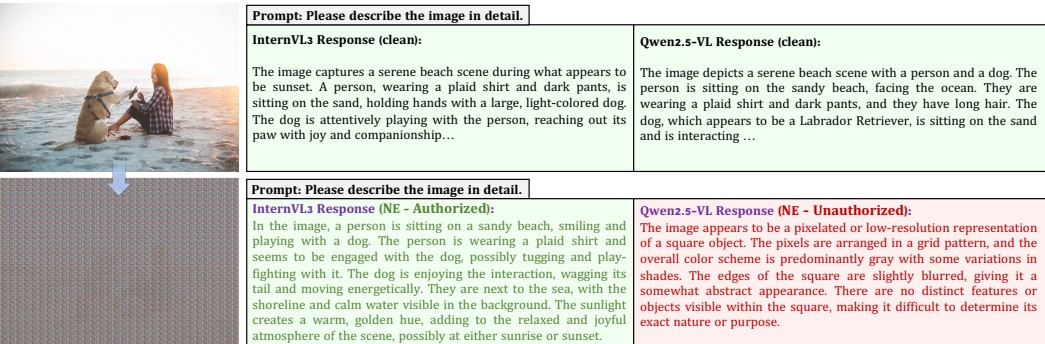

Figure 2: Illustrative visualization of effective on data authorization on VLM.

**VLMs**. Beyond standard backbones, we extend NE to state-of-the-art vision-language models. We evaluate `InternVL3` (authorized) and `Qwen2.5-VL` (unauthorized) on the comprehensive benchmark MMBench (Liu et al., 2024b), across capability dimensions AR, CP, FP-C, FP-S, LR, and RR.

Table 4: VLMs on MMBench.

| | InternVL3-1B (authorized) | | | | | | Qwen2.5-VL-3B (unauthorized) | | | | | |
|---|---|---|---|---|---|---|---|---|---|---|---|---|
| | Overall | AR | CP | FP-C | FP-S | LR | RR | Overall | AR | CP | FP-C | FP-S | LR | RR |
| Baseline | 72.7% | 77.4% | 82.4% | 58.7% | 79.4% | 50.3% | 66.8% | 78.8% | 81.3% | 83.1% | 69.2% | 84.9% | 66.5% | 75.4% |
| NE (Ours) | 72.6% | 77.8% | 82.0% | 58.3% | 78.9% | 50.9% | 67.3% | 18.3% | 29.2% | 15.6% | 17.0% | 12.8% | 17.3% | 22.3% |

As shown in Table 4, the authorized model remains essentially unchanged, while the unauthorized model is consistently low and remains unusable across settings. An illustrative example is shown in Figure 2 (see more in Appendix 7), recoded inputs preserve authorized performance and suppress unauthorized utility, where the unauthorized model sees it as completely random noise pixels.

**NE for Skin-lesion Classification**. NEs are naturally applicable in high-stakes, privacy-sensitive domains, since the construction is task-agnostic and only depends on the first-layer representation of the model. To illustrate this in a realistic healthcare scenario, we additionally evaluate NE on the HAM10000 (Nagabu, 2024) dermoscopic skin-lesion dataset (HuggingFace `Nagabu/HAM10000`). A ViT fine-tuned on this task reaches 99.2% accuracy on clean images; after applying NEs, the authorized ViT still achieves 99.2% accuracy. In contrast, An unauthorized ResNet-50, likewise fine-tuned on the dataset, performs well on clean images but collapses to 0.0% accuracy on the same NE-encoded inputs.[2]

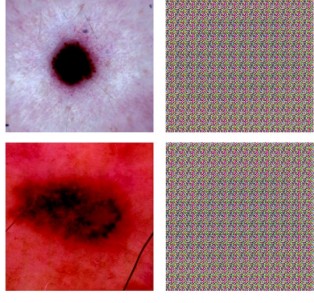

**Runtime Analysis**. NE generation is lightweight and practical for both real-time inference and large-scale deployment. The only non-trivial step is a one-time SVD of the first linear map $W \in \mathbb{R}^{m \times n}$ to obtain the insensitive subspace, done once per authorized model. After that, producing an NE for a new input only requires sampling $z \in \mathbb{R}^n$, computing a small matrix-vector product

---

[2]The test set is heavily imbalanced with more than 80% positives.

$\delta = Vz$, and adding $\delta$ to $x$. For a standard $224 \times 224$ RGB image with a ResNet-50 front (first layer $W \in \mathbb{R}^{64 \times 147}$), we measure an average of 0.47 ms for the SVD + basis construction (one-time) and 0.11 ms per-image for NE encoding, which is negligible compared to a single forward pass.

**Robustness of NEs**. The robustness of NEs to common distortions follows directly from the construction: for the authorized model, the recoding lives in low-sensitivity directions of the first layer and is effectively neutralized after that layer, so moderate changes to the input (resize, crop, mild compression, blur) do not selectively disrupt authorization. On ViT-B fine-tuned on CIFAR-10, random crops that preserve most of the object (crop ratios 10%-40%) keep NE(auth) accuracy within about 1-3 percentage points of the clean baseline, while NE(unauth) on non-target models stays near random-guess level. Under lossy JPEG compression, as quality decreases from 95 to 75, clean accuracy drops slightly (79.1% to 77.7%), NE(auth) decreases from 77.0% to 68.4%, and NE(unauth) again remains around 10%, so realistic compression modestly degrades the underlying task but does not restore unauthorized utility. In the VLM setting, where preprocessing pipelines are more complex (see Appendix D.3), we observe the same pattern: the authorized InternVL3 retains essentially its baseline performance on NE-encoded inputs, whereas the unauthorized Qwen2.5-VL collapses across all capability dimensions (Table 4).

We also study stronger attacks beyond fixed preprocessing. For super-resolution reconstruction, we apply SR-ResNet (Li et al., 2018) to VDSR (Vedaldi & Lenc, 2015) in both a black-box setting (trained on perturbed inputs and targets) and a white-box setting (trained with clean targets). In both cases, the reconstructed images remain visually uninformative, improve SNR by at most $0.6$ dB, and fail to recover downstream classification accuracy, while authorized performance on NEs is essentially unchanged (Appendix D.4).

Finally, we consider a learned adaptation attack in which unauthorized models are trained directly on NE-encoded data. Using the first-layer weights of a ViT-B (98.7% accuracy on NEs) to construct the NE basis on CIFAR-10, we generate NEs and fine-tune four unauthorized backbones, ResNet-50, SwinV2-T, DeiT-B, and MambaVision-T, on pairs $(\tilde{x}, y)$ for 3 epochs (batch size 64, learning rate $5 \times 10^{-4}$, full-parameter tuning). After training, their accuracies on NEs remain near chance (10.8%, 10.0%, 10.8%, and 10.1%, respectively), while the authorized ViT-B stays at 98.7% (full details in Appendix D.5). These results indicate that NE not only tolerates realistic preprocessing pipelines but also resists substantially stronger reconstruction and adaptation attempts: the authorized model retains utility on NEs, whereas unauthorized models fail to recover meaningful performance.

## 6 RELATED WORK

**Training-time Defenses.** Anti-learnability perturbs released data so standard training fails while human perception is preserved (Ye & Wang, 2024; Wang et al., 2025a). Non-transferable training modifies objectives or parameters to suppress transfer in designated domains (Wang et al., 2022; Hong et al., 2025). Both act during training and do not control inference once the content is public.

**Algorithmic Authorization.** This line of work binds data utility to a chosen algorithmic class. AlgoSpec applies polynomial approximation so that only a designated algorithm family recovers accuracy on transformed inputs (Liu et al., 2024a). In practice, it is only limited to simple classifiers such as Naive Bayes (Rish et al., 2001) and does not extend to neural networks.

**Differential Privacy and Encrypted Inference.** DP limits training-time leakage from individual examples into the learned model (Dwork et al., 2014), but it does not control who can run inference on public inputs. Fully homomorphic encryption enables encrypted inference with strong confidentiality (Gentry, 2009) but incurs substantial latency and memory overheads (Ribeiro et al., 2015; Meftah et al., 2021). We instead pursue lightweight, practical, model-specific authorization.

## 7 CONCLUSION

We presented non-transferable examples (NEs), a lightweight mechanism that preserves data utility for an authorized model while denying it to unauthorized ones. Our theory guarantees utility retention and quantifies degradation, and our experiments confirm robustness across diverse architectures and modalities. Together, these results show that NEs offer a practical path to model-level usage control, ensuring data serves its intended purpose without enabling misuse.

## ETHICS STATEMENT

This research does not involve human subjects, personally identifiable information, or sensitive datasets. Our method, non-transferable examples (NEs), is developed to mitigate unauthorized model use while preserving utility for intended applications. While usage-control mechanisms naturally intersect with broader discussions of openness and accessibility, our focus is on preventing misuse and supporting responsible AI practice. This work is intended to complement emerging regulatory and community standards for trustworthy AI.

## REPRODUCIBILITY STATEMENT

We provide full details of our method, including theoretical foundations, algorithms, and hyperparameters. Experiments are run on standard datasets (CIFAR-10, ImageNet, MMBench) and widely used architectures (ResNet, ViT, Swin, DeiT, MambaVision, Qwen2.5-VL, InternVL3). Code, preprocessing scripts, and a demo showcasing non-transferable examples on representative models will be made available at: https://github.com/model-specific/non-transferable-examples.git.

## ACKNOWLEDGMENT OF LLM USE

The role of the LLM in this work was limited to polishing text for grammar and readability. The intellectual and technical contributions are solely those of the authors.

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

APPENDIX

## A  THREAT MODEL ASSUMPTIONS

Our approach assumes that the defender has white-box access to the authorized model's first-layer weights. This is reasonable in *server-side deployments*, where providers retain full control over the inference stack (*e.g.,* Google AutoML, AWS SageMaker, Microsoft Azure ML). Similar assumptions are standard in adversarial ML defenses such as randomized smoothing (Cohen et al., 2019) and spectral regularization (Zhang et al., 2020), which likewise rely on internal model access. This is also aligned with regulatory expectations (*e.g.,* EU AI Act, NIST AI RMF), where model owners are held responsible for ensuring compliance and thus are expected to enforce usage control internally.

We further assume that adversaries cannot reliably estimate the insensitivity subspace of the authorized model. Recovering such information would require either access to first-layer weights or large-scale probing with structured queries. Prior work on model extraction (Tramèr et al., 2016; Ma et al., 2024a) and property inference (Ganju et al., 2018) demonstrates that approximating hidden representations typically requires millions of queries, which is unrealistic under commercial API rate limits and cost constraints (Bai et al., 2013; Yan et al., 2024). Moreover, because our perturbations lie in directions of negligible sensitivity for the authorized model, they are indistinguishable from natural input variance to unauthorized models, making subspace approximation even harder in practice.

## B  SUPPLEMENTARY PRELIMINARIES

### B.1  NEURAL NETWORKS

Neural networks are a class of machine learning models inspired by the structure and function of the human brain. They consist of interconnected layers of neurons (nodes), which process input data to produce an output. Each neuron applies a mathematical function to its inputs, typically involving weights and biases, along with an activation function that is adjusted during training to minimize the difference between the predicted and actual outputs. A neural network is typically structured as a series of stacked linear transformations, followed by nonlinear activation functions. Formally, a neural network can be represented as a function $f : \mathbb{R}^n \to \mathbb{R}^m$, where $n$ is the number of input features and $m$ is the number of output classes or values. Here, we take the feedback neural network as an example (*e.g.,* fully connected or convolutional neural networks). Because convolutional operations are linear transformations that can be unfolded into fully connected layers, we focus on fully connected neural networks. Further, the bias terms $b^{(i)}$ can be taken as part of the weight matrix $W^{(i)}$ of the $i$-th layer by appending a constant input of $1$ to the input vector $x$, so we can simplify and rewrite the notation for theoretical convenience as

$$y = f(x) = \phi(W^{(n)} \cdot \phi(W^{(n-1)} \cdot \cdots \phi(W^{(2)} \cdot \phi(W^{(1)}x)) \cdots)),$$

where $\phi$ is the activation function, which introduces non-linearity into the model. Common activation functions include the rectified linear unit (ReLU), sigmoid, and hyperbolic tangent (tanh).

### B.2  EIGENDECOMPOSITION AND PRINCIPAL COMPONENT ANALYSIS

A neural network is trained on a dataset to learn the underlying patterns and relationships in the data. Principal component analysis (PCA) is a technique used to analyze the learned representations of data by transforming it into a new coordinate system. It performs the eigendecomposition of the data's covariance matrix to identify the directions (principal components) that maximize the variance in the data. It is a linear transformation that projects the data onto a lower-dimensional subspace defined by the principal components, which are determined by the eigen decomposition of the covariance matrix of the data. The eigen vectors of the covariance matrix represent the directions of maximum variance, while the eigenvalues indicate the amount of variance along those directions.

Before giving the formal definition of PCA, we first introduce the classical eigendecomposition of a real-valued matrix without proof for simplicity.

**Definition 2** (Eigendecomposition). *The eigendecomposition of a square matrix $C \in \mathbb{R}^{n \times n}$ is a factorization of the form $C = V \Lambda V^{\top}$, where $V \in \mathbb{R}^{n \times n}$ is a orthogonal matrix whose columns are the eigen vectors of $C$, and $\Lambda \in \mathbb{R}^{n \times n}$ is a diagonal matrix whose diagonal entries are the eigenvalues of $C$. The eigenvalues are the scalars $\lambda_i$ such that $Cv_i = \lambda_i v_i$, where $v_i$ is the $i$-th eigen vector of $C$.*

In this work, the eigenvalues are arranged in ascending order along the diagonal of $\Lambda$ by default. Geometrically, $V^{\top}$ represents the rotation of the coordinate system, $\Lambda$ scales the axes of the new coordinate system according to the eigenvalues, and $V$ rotates the data back to the original coordinate system. Formally, given a dataset represented as a matrix $x \in \mathbb{R}^n$, where $n$ is the input dimension, PCA can be performed as follows.

**Definition 3** (Principal Component Analysis (PCA)). *The covariance matrix is $C = \mathbb{E}[x^{\top}x]$, where $\mathbb{E}$ denotes the expectation operator. The eigen decomposition of the covariance matrix is given by $C = V_{pca} \Lambda V_{pca}^{\top}$, where $V_{pca}$ is the matrix of eigen vectors (column vectors) and $\Lambda$ is the diagonal matrix of eigenvalues. The principal components are the columns of the matrix $V_{pca}$, and the projection of the data onto the principal components is given by $V_{pca}^{\top}x$.*

PCA is widely used for dimensionality reduction by selecting the top $k$ principal components, where $k$ is the desired number of dimensions. However, here we focus on analyzing the learned representations of the data using all principal components.

**PCA in neural networks**. PCA can be applied as a valuable tool to analyze the learned representations of data in neural networks. It describes the patterns and relationships of data input dimensions in the learned representations, *e.g.,* several input dimensions are enough to represent the data for a specific classification task.

### B.3 SINGULAR VALUE DECOMPOSITION

Since the trainable parameters of a neural network comprise the weight matrices and biases, we introduce the singular value decomposition (SVD) to decompose these matrices for further analysis. Considering the merged weight matrix in the previous Appendix B.1, we focus on the weight matrix $W$ without the bias terms for simplicity. The SVD of a matrix is defined as the following lemma, where we only consider the case of real-valued matrices and only provide this typical result without proof for simplicity.

**Lemma 1** (Singular Value Decomposition (SVD)). *For any matrix $W \in \mathbb{R}^{m \times n}$, there exist orthogonal matrices $U \in \mathbb{R}^{m \times m}$ and $V \in \mathbb{R}^{n \times n}$, and a diagonal matrix $\Sigma \in \mathbb{R}^{m \times n}$ such that $W = U \Sigma V^{\top}$. The diagonal entries of $\Sigma$ are the singular values of $W$, and the columns of $U$ and $V$ are the left and right singular vectors, respectively.*

The singular values are non-negative and arranged in ascending order along the diagonal of $\Sigma$. Geometrically, the SVD decomposes the matrix $W$ into three components: $U$ represents the rotation of the input space, $\Sigma$ scales the axes according to the singular values, and $V^{\top}$ represents the rotation of the output space. The diagonal entries of $\Sigma$ are the singular values, which indicate the importance of each corresponding singular vector in the decomposition. The smaller the singular value, the less important the corresponding singular vector is in representing the original matrix.

**SVD in neural networks**. In neural networks, applying SVD to a layer's weight matrix reveals how inputs are prioritized. The singular vectors indicate influential directions in the input space, and the singular values quantify their relative strength. For the first layer, this analysis is closely related to PCA of the input data: dominant data components often align with the layer's most responsive input directions (up to whitening and scaling). Because the first layer extracts features from raw inputs, SVD offers a clear view of how input dimensions are transformed into learned representations.

### B.4 NULLSPACE

When we take a matrix as a linear transformation, the nullspace of a matrix refers to the set of vectors that are mapped to the zero vector by that matrix. Formally, we have the following definition.

**Definition 4** (Nullspace). *The nullspace of a matrix $W \in \mathbb{R}^{m \times n}$, denoted as $\mathrm{Null}(W)$, is defined as $\mathrm{Null}(W) = \{x \in \mathbb{R}^n \mid Wx = 0\}$.*

The nullspace of a matrix $W \in \mathbb{R}^{m \times n}$ is the subspace of $\mathbb{R}^n$ containing all vectors that $W$ maps to the zero vector. Its dimension is the *nullity* of $W$, and by the rank-nullity theorem the sum of the rank and nullity equals $n$, the number of columns.

**Nullspace in neural networks**. For a linear layer with weight matrix $W$, the nullspace consists of input directions that produce no change at that layer's output (*i.e.,* directions the layer is effectively insensitive to). In a multilayer network, such directions are suppressed before subsequent processing and thus have negligible downstream influence. This is especially intuitive in early vision layers, where convolutional filters emphasize specific spatial-frequency patterns; inputs orthogonal to those patterns lie (approximately) in low-response or null directions.

### B.5 CONVOLUTION

This section clarifies that the convolution operation, which is commonly used in famous convolutional neural networks (CNNs), is a linear transformation that can be represented as a matrix multiplication. We first define the convolution as follows. We refer to a multi-dimensional matrix as a tensor, and omit commonly used arguments in the current code implementation, such as batch size, padding, stride, dilation, and group, for simplicity, as they can be easily extended by adding extra zero dimensions to the input tensor or kernel.

**Definition 5** (Convolution). *The convolution takes inputs including an input tensor $X \in \mathbb{R}^{c_1 \times h \times w}$ and a kernel (filter) $K \in \mathbb{R}^{c_1 \times c_2 \times k_h \times k_w}$ with the kernel bias $b \in \mathbb{R}^{c_2}$, where $c_1$ and $c_2$ are the number of channels in the input tensor and kernel, respectively, and $h$, $w$, $k_h$, and $k_w$ are the height and width of the input tensor and kernel. The convolution operation outputs a tensor $Y \in \mathbb{R}^{c_2 \times h' \times w'}$, where $h'$ and $w'$ are the height and width of the output tensor, and each element of the output tensor is computed as follows,*

$$Y_{c_2,i,h_i',w_i'} = \sum_{c_1,j=1}^{c_1} \sum_{h_j'=1}^{k_h} \sum_{w_j'=1}^{k_w} K_{c_1,j,c_2,i,h_j',w_j'} \cdot X_{c_1,j,h_i'+h_j'-1,w_i'+w_j'-1} + b_{c_2,i},$$

**Convolution as matrix multiplication**. The convolution operation can be represented as a matrix multiplication by *unfolding* the input tensor into a matrix and the kernel into a matrix and then performing the matrix multiplication. The resulting matrix can be *folded* back into a tensor to obtain the output of the convolution operation. Due to the application of convolution in image processing, the unfolding and folding operations are also known as *im2col* and *col2im*, respectively.

**Lemma 2** (Convolution as Matrix Multiplication). *The convolution operation can be represented as a matrix multiplication by unfolding the input tensor $x$ into a matrix $X' \in \mathbb{R}^{c_1 k_h k_w \times h' w'}$ and the kernel $k$ into a matrix $K' \in \mathbb{R}^{c_2 \times c_1 k_h k_w}$, where $h'$ and $w'$ are the height and width of the output tensor. By processing matrix multiplication $Y' = K'X' + b$, we can obtain the output tensor $Y' \in \mathbb{R}^{c_2 \times h' w'}$ and then fold it back into a tensor $Y \in \mathbb{R}^{c_2 \times h' \times w'}$.*

*Proof.* We provide only a sketch of the proof here, which can be easily verified by the definition of the convolution operation. For the input tensor $X$ and its unfolded matrix $X'$, we only need to extract each local patch of the input tensor that corresponds to the kernel size and reshape it into a column vector, which is called a Toeplitz matrix. The kernel $K$ can be reshaped into a matrix $K'$ by stacking the kernel channels and kernel dimensions into a single dimension. The output tensor $Y$ can be obtained by performing the matrix multiplication $Y' = K'X'$. Finally, we reshape the output matrix $Y'$ back into a tensor $Y$ by folding it into the original shape of the output tensor. Note that in this simple scenario, we only need to reshape the output matrix $Y'$ into a tensor $Y$. However, a more complex scenario requires folding the output matrix. $\square$

### B.6 TOKEN EMBEDDING

Token embedding is a technique used in natural language processing (NLP) to convert discrete tokens (such as words or subwords) into continuous vector representations (Feng et al., 2024; Devlin et al., 2019). This is essential for enabling neural networks to process text data, as neural networks typically operate on continuous numerical data. Token embedding maps each token to a high-dimensional vector space, where similar tokens are represented by vectors that are close to

each other in that space. In the context of neural networks, token embedding is taken as a linear transformation that maps the input tokens to a continuous vector space. Formally, we define the token embedding as follows.

**Definition 6** (Token Embedding). *The token embedding is a linear transformation that maps a discrete token $t \in \mathbb{R}^d$ to a continuous vector representation $e \in \mathbb{R}^m$ using a weight matrix $W \in \mathbb{R}^{m \times d}$ and a bias vector $b \in \mathbb{R}^m$. The token embedding is defined as $e = Wt + b$, where $W$ is the weight matrix that maps the input token to the continuous vector space, and $b$ is the bias vector that shifts the output vector.*

## C  SUPPLEMENTARY ASSUMPTIONS

This section summarizes assumptions made in the paper, which are crucial for understanding the theoretical framework and implications of the results. We refer to the *spectral distribution* of a matrix as its eigenvalues or singular values, depending on the context.

### C.1  SPECTRAL FLATNESS OF DATA REPRESENTATIONS

The spectral distribution of a matrix is said to be *flat* if the eigenvalues or singular values are uniformly distributed across a certain range, *i.e.,* several eigenvalues or singular values are close to each other in magnitude. Such property is often satisfied in high-dimensional practical applications. For instance, high-resolution images in computer vision introduce a flat spectral distribution because adjacent pixels are highly correlated; similarly, word embeddings in natural language processing exhibit a flat spectral distribution, as words are often used in similar contexts.

Formally, we make the following assumption about the spectral distribution of the data representations.

**Assumption 1** (Spectral Flatness of Data Representations). *Given a dataset for a specific task, the PCA of the data has a flat spectral distribution,* i.e., *there are several eigenvalues, $\sigma_k$, $\sigma_{k+1}$, $\sigma_{k+2}$, ..., that are close to each other in magnitude,*

$$\sigma_1 \leq \sigma_2 \leq \cdots \leq \sigma_k \lessapprox \sigma_{k+1} \lessapprox \sigma_{k+2} \lessapprox \cdots \leq \sigma_n,$$

*where $\lessapprox$ denotes that the values are close to each other in magnitude, and $\sigma_i$ are the singular values of the data matrix.*

This assumption holds broadly in practice: *e.g.,* in vision and language, where data (or early feature) representations exhibit relatively flat spectra with clustered eigenvalues. In many real datasets, multiple flats (plateaus) appear, each reflecting a distinct group of features or patterns. We empirically verify this in Figure 3: for each model, we report the minimum number of singular components needed to capture 95% (P95) and 99% (P99) of total spectral energy. ResNet-50 (He et al., 2016) reaches 95.4% with 23 components (P95@23) and 99.0% with 32 (P99@32). ViT-Base (Dosovitskiy et al., 2021), despite a higher-dimensional patch projection, requires 90 and 106 components to attain 95.4% and 99.0%, respectively.

### C.2  ALIGNMENT OF SINGULAR VECTORS AND PRINCIPAL COMPONENTS

Training a neural network amounts to optimizing parameters to minimize a task loss. In this process, the right singular vectors of the first-layer weight matrix emphasize directions that are most predictive for the task, whereas the principal components of input (or early-feature) representations summarize dominant data variability. Empirically, these two families of directions often exhibit notable alignment in practical settings (Hacohen & Weinshall, 2022). This section formalizes that observation by relating first-layer right singular vectors to the principal components of data representations, and proceeds under the following alignment assumption.

**Assumption 2** (Alignment of Singular Vectors and Principal Components). *Given a dataset and a neural network trained on this dataset, the first layer's weight matrix is denoted as $W \in \mathbb{R}^{m \times n}$, where $n$ is the number of input dimensions and $m$ is the number of output dimensions, and the covariance matrix of the data representations is denoted as $C \in \mathbb{R}^{n \times n}$. Let $U\Sigma V_{svd}^T$ be the singular value decomposition (SVD) of the first layer's weight matrix $W$, where $U \in \mathbb{R}^{m \times m}$ is the left*

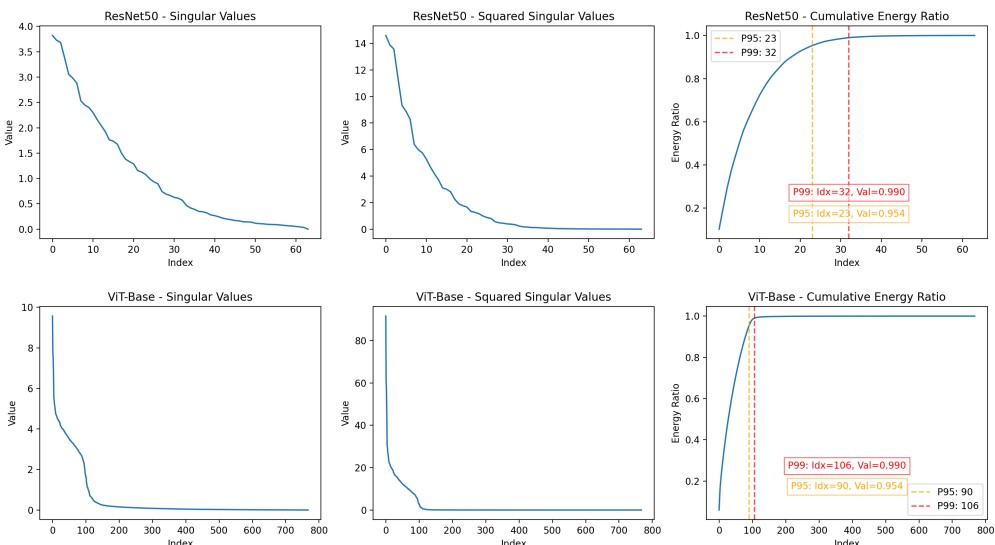

Figure 3: The spectral structure of the first-layer weight matrices in ResNet-50 and ViT-Base via singular value decomposition.

*singular vector matrix, $\Sigma \in \mathbb{R}^{m \times n}$ is the diagonal matrix of singular values, and $V_{svd} \in \mathbb{R}^{n \times n}$ is the right singular vector matrix. Let $V_{pca} \Lambda V_{pca}^T$ be the eigendecomposition of the covariance matrix $C$, where $V_{pca} \in \mathbb{R}^{n \times n}$ is the eigen vector matrix and $\Lambda \in \mathbb{R}^{n \times n}$ is the diagonal matrix of eigenvalues. The assumption states that the right singular vectors of the first layer's weight matrix are aligned with the principal components of the data representations, that is*

$$V_{svd} = V_{pca}.$$

This assumption is satisfied in many practical applications, where the right singular vectors of the first layer weight matrix are aligned with the principal components of the data representations. The alignment between the right singular vectors and the principal components is crucial for the neural network to learn the most important features from the data for the task. In some special models, such as linear models, the alignment is exact, *i.e.*, $V_{svd} = V_{pca}$.

Table 5: Numerical experiment on first-layer singular-value.

|  | Max | Min | Mean | Median |
|---|---|---|---|---|
| ResNet-50 | 3.8203 | $7.77 \times 10^{-8}$ | 1.0242 | 0.5904 |
| ViT-Base | 9.5686 | $3.68 \times 10^{-5}$ | 0.5666 | 0.0482 |

### C.3 DEEPER LAYER ANALYSIS AND ARCHITECTURE COVERAGE

Our analysis explicitly controls the perturbation at the first linear map rather than relying on later layers to implicitly preserve the desired property. For directions in the exact null space of the first linear map $W$, we have $W\delta = 0$, so the first-layer features are unchanged and $f^\star(x + \delta) = f^\star(x)$ follows immediately. For near-null directions in our $\tau$-insensitive subspace, Eq. 4 gives a quantitative bound on the first-layer change, $\|W\tilde{x} - Wx\|_2 \leq \varepsilon(\tau, \lambda)$, so the input fed to all subsequent layers is only slightly and boundedly perturbed.

This upper bound extends naturally to the whole network by writing $f^\star = g_L \circ \cdots \circ g_1$ with finite operator or Lipschitz norms $L_\ell$ and applying a simple induction over layers. This guarantees that the deviation remains bounded at every depth and thus yields the target-retention guarantee for the entire model. In practice, activation nonlinearities such as ReLU (zeroing negatives) and saturating sigmoids or `tanh` further damp the propagation of such small perturbations, so near-null directions tend to attenuate quickly across depth.

Building on this compositional view, the same reasoning applies in a model-agnostic way to concrete architectures such as ResNets. We operate on the very first mapping from pixels to features (*e.g.,* applying `nn.unfold()` to the $7 \times 7$ convolution in ResNet) and select perturbations that keep this

mapping unchanged or only slightly changed for the authorized model. Formally, Eq. 4 bounds the first-layer deviation by a small value,

$$\|W\tilde{x} - Wx\|_2 \le \varepsilon(\tau, \lambda).$$

In a residual block of the form $h_{l+1} = h_l + F(h_l)$, if we write $\Delta_l = \tilde{h}_l - h_l$, then

$$\Delta_{l+1} = \Delta_l + \big(F(h_l + \Delta_l) - F(h_l)\big).$$

Since $F$ is a composition of linear or convolutional layers and activations, it has a finite Lipschitz constant $L_F$, giving

$$\|\Delta_{l+1}\|_2 \le \|\Delta_l\|_2 + \|F(h_l + \Delta_l) - F(h_l)\|_2 \le (1 + L_F)\,\|\Delta_l\|_2.$$

Chaining this over layers yields a quantifiable bound on the deviation at any depth that is ultimately tied to the small first-layer bound from Eq. (4). With our empirically chosen hyperparameters in the insensitive subspace (Appendix C.2), these deviations for the authorized model remain in a small, controllable range that stays well inside the decision boundary margin, which matches Figure 1 where authorized accuracy stays essentially flat across perturbation levels. For unauthorized models, the same perturbation is no longer near-null in their first mapping, so their early features are already significantly distorted, and residual blocks then propagate (and can amplify) this mismatch, leading to the strong non-transferability we observe.

To further validate this picture, we report layer-wise deviations for ResNet-50 under different NE strengths (PSNR from 40 to 0 dB). We index layers by forward operations (including convolution, BatchNorm, ReLU, and downsample $1 \times 1$ convolutions) and record the norm every 40 operations, denoted L0L6. Across all PSNR levels, the perturbation injected at the input stays small throughout the network and grows smoothly as we lower PSNR, with no exploding behavior in deeper layers. For the authorized model, even at aggressive perturbation levels (0 dB) the deviations remain well controlled at all checkpoints, supporting our claim that near-null perturbations stay close to zero across depth, consistent with the flat authorized-accuracy curves in Figure 1.

Table 6: Layer-wise deviation norms for ResNet-50 under different NE strengths (PSNR in dB). L0L6 index checkpoints every 40 forward operations.

| PSNR (dB) | L0 | L1 | L2 | L3 | L4 | L5 | L6 |
|---|---|---|---|---|---|---|---|
| 40 | 0.0032 | 0.0097 | 0.0014 | 0.0062 | 0.0043 | 0.0210 | 0.0008 |
| 35 | 0.0035 | 0.0107 | 0.0017 | 0.0072 | 0.0050 | 0.0247 | 0.0010 |
| 30 | 0.0071 | 0.0200 | 0.0031 | 0.0146 | 0.0102 | 0.0498 | 0.0023 |
| 25 | 0.0134 | 0.0331 | 0.0052 | 0.0272 | 0.0184 | 0.0859 | 0.0040 |
| 20 | 0.0112 | 0.0367 | 0.0068 | 0.0282 | 0.0207 | 0.1016 | 0.0046 |
| 15 | 0.0240 | 0.0680 | 0.0104 | 0.0499 | 0.0349 | 0.1672 | 0.0079 |
| 10 | 0.0513 | 0.1731 | 0.0293 | 0.1108 | 0.0849 | 0.4344 | 0.0192 |
| 5 | 0.0754 | 0.3013 | 0.0501 | 0.1506 | 0.1239 | 0.6520 | 0.0264 |
| 0 | 0.1916 | 0.5643 | 0.0994 | 0.4031 | 0.2991 | 1.5062 | 0.0674 |

## D  SUPPLEMENTARY CLARIFICATIONS, EXPERIMENTS, AND RESULTS

This section provides additional discussion and experimental results. All experiments use Python 3.12.3, PyTorch 2.3.0, and Transformers 4.44.2 (CUDA 12.3) on a workstation with an AMD Ryzen Threadripper PRO 5965WX (24 cores), 256 GB RAM, and two NVIDIA RTX A6000 GPUs.

Table 7 presents the performance of different models on the GLUE benchmark to demonstrate that our approach is generic. Figure 4 illustrates the effect of perturbation strength on an example image.

### D.1  CLARIFICATION ON VISUAL QUALITY

During implementation we think that different applications care about visual quality in different ways, and NES can enforce model-specificity while accommodating both. Our experiments (Figure 1, with a closer view in Figure 4) show that as we gradually increase the perturbation strength, the authorized model's accuracy stays essentially unchanged while unauthorized models significantly collapse, providing a "useful range" to choose from.

- When humans still need to see the image after encoding (for example, internal dashboards or research workflows where people occasionally inspect samples), one can use milder NES so that images remain visually clear while still being non-transferable to other models.

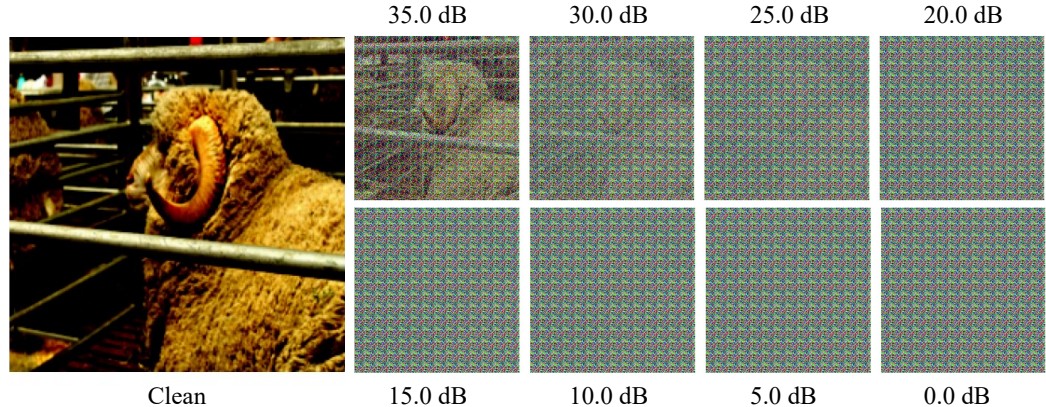

Figure 4: Effect of perturbation strength (PSNR, dB). Visual examples across increasing strength. Authorized models remain stable even at 10dB; at 0dB, ResNet-50 on ImageNet loses only 0.1% accuracy.

- When only the model's output matters and humans never look at the encoded data (for example, server-side MLaaS or external processing of sensitive medical/industrial imagery), stronger NEs are acceptable: the encoded image may look heavily distorted to humans but is still correctly processed by the authorized model $f^\star$.

This flexibility lets practitioners pick a perturbation level that matches the visual requirements of their application, while still enforcing model-specific authorization on $\tilde{x}$.

## D.2 MULTI-CLIENTS USAGE

If there are multiple client models, the data owner can handle them in two natural ways. 1) The data owner can generate a separate NE version for each model: for each $f_i$, estimate its insensitivity subspace from the first layer, define $\mathcal{T}_{f_i}$, and produce $\tilde{x}_i = \mathcal{T}_{f_i}(x)$; each client then receives the version tied to its own model. Since NE construction has negligible overhead (see Q5), producing several per-model encodings is practical for both real-time inference and large-scale deployment. 2) When a small group of models should share the same encoded data, one can approximate a joint insensitive subspace (*e.g.,* by intersecting or averaging their low-sensitivity directions) and build NEs that preserve all of them simultaneously. This multi-model construction is conceptually straightforward but may involve a different trade-off between authorized retention and non-target degradation, which we leave as an interesting direction for future empirical study. We will briefly discuss these two multi-model options in the revision.

Table 7: Performance across GLUE benchmark for different models.

| | BERT-base | | | | | | | RoBERTa-base | | | | | | |
|---|---|---|---|---|---|---|---|---|---|---|---|---|---|---|
| | CoLA | MNLI | QNLI | QQP | RTE | SST2 | STSB | CoLA | MNLI | QNLI | QQP | RTE | SST2 | STSB |
| Baseline | 54.2 | 83.4 | 90.5 | 90.1 | 60.3 | 91.6 | 87.1 | 53.8 | 87.7 | 92.8 | 90.9 | 66.1 | 94.5 | 87.5 |
| NE (Ours) | 54.5 | 82.9 | 89.5 | 89.6 | 60.3 | 89.6 | 86.9 | 55.5 | 87.5 | 92.4 | 90.8 | 65.3 | 94.6 | 87.2 |
| Unauthorized | 33.4 | 31.8 | 50.5 | 36.8 | 47.3 | 50.9 | 60.2 | 37.1 | 35.4 | 49.5 | 63.2 | 52.7 | 49.1 | 63.5 |

## D.3 ROBUSTNESS AGAINST PREPROCESSING IN VLMS

Vision-language models apply multi-stage, model-specific preprocessing that can scramble input space recoding before it reaches early features, which makes inference time usage control difficult. In practice images are decoded to RGB and converted to floats, resized to model specific canvases such as 448, 512, 896 or 1024 with aspect ratio preserved and letterbox padding, optionally center or random cropped, normalized with per channel means and standard deviations similar to CLIP or EVA, partitioned into patches or tiles to form visual tokens with padding aligned to stride and patch size, projected into the language model embedding space and augmented with resolution dependent positional encodings, with occasional multi image packing and implementation dependent interpolation and JPEG rounding. Although InternVL3 and Qwen2.5 VL differ in exact choices, they follow this general pattern. Despite these complications, our experiments show that NE remains robust on MMBench across AR, CP, FP C, FP S, LR, and RR, where authorized performance for InternVL3

is essentially unchanged and unauthorized utility for Qwen2.5 VL remains low as summarized in Table 4 and illustrated in Figure 2.

## D.4    Reconstruction Attack

Table 8 summarizes super-resolution reconstruction attempts on VDSR using SR-ResNet in black-box (Noise2Noise) and white-box (Noise2Clean) settings. As a sanity check, standard Gaussian noise is largely removable (33.5–35.9dB after SR-ResNet). In contrast, NE resists recovery: reconstructions stay near the input level (about 10–11dB) in both settings, including the strongest white-box variant with all layers. Thus, recoding remains effectively non-invertible for attackers while leaving authorized performance essentially unchanged.

Table 8: PSNR (dB) on VDSR under super-resolution reconstruction attempts.

| Method | $\tilde{x}$ | SR-ResNet (black box)[1] | SR-ResNet (white box)[2] |
|---|---|---|---|
| Gaussian | 15.8 | 33.5 | 35.9 |
| NE (Ours) | 10.2 | 10.5 | 10.9 |
| NE (Ours)[3] | 10.7 | 10.8 | 11.1 |

[1] Noise2Noise: train on recoded → recoded pairs.
[2] Noise2Clean: train on recoded → clean pairs.
[3] Attacker has the white box access to the model.

## D.5    Learned Adaptation Attacks

In addition to fixed preprocessing and reconstruction, we also consider a stronger "learned adaptation" attacker who is allowed to train directly on NE-encoded data. The question is whether an unauthorized model can be fine-tuned to absorb the NE perturbation and recover high accuracy on encoded inputs.

**Setup**. We work on CIFAR-10 with its standard train/test split. We first fine-tune a ViT-B model on clean CIFAR-10 and use its first-layer weights (which achieve 98.7% accuracy on NE-encoded inputs in the authorized setting) to construct the NE basis as described in Section 3. We then generate NE-encoded versions of the training and test images and treat these as the attacker's dataset.

As unauthorized models, we consider ResNet-50, SwinV2-T, DeiT-B, and MambaVision-T. Each unauthorized model is trained from its standard initialization on the NE-encoded training set using the original CIFAR-10 labels. We fine-tune all parameters for 3 epochs with batch size 64 and learning rate $5 \times 10^{-4}$. The attacker thus has access to $(\tilde{x}, y)$ pairs and is allowed to adapt the entire model to the NE distribution.

**Results**. After training, we evaluate all models on the NE-encoded test set. As summarized in Table 9, all unauthorized backbones remain near chance-level accuracy on NEs (around 10% for a 10-class task), while the authorized ViT-B maintains its 98.7% accuracy on the same NE-encoded inputs.

Table 9: Accuracy of unauthorized models trained directly on NE-encoded CIFAR-10 (3 epochs, full fine-tuning). The authorized ViT-B remains at 98.7% accuracy on NEs.

| Model | Accuracy on NEs |
|---|---|
| ResNet-50 | 10.8% |
| SwinV2-T | 10.0% |
| DeiT-B | 10.8% |
| MambaVision-T | 10.1% |

These results show that even when an attacker is allowed to fine-tune strong unauthorized architectures directly on NE-encoded data, they fail to recover meaningful performance and remain at chance, whereas the authorized ViT-B continues to perform well on NEs.

## E    Discussion of Limitations

Our setting grants the defender white-box access to $f^\star$ and a probe source to estimate an insensitivity subspace used for recoding. Against a *method-aware* or *parameter-aware* adversary (GA/TA/AA in Section 2.2), the perturbation itself becomes an attack surface. If the confining subspace (or a close approximation) is recovered, an input-side projector that reweights toward principal directions can

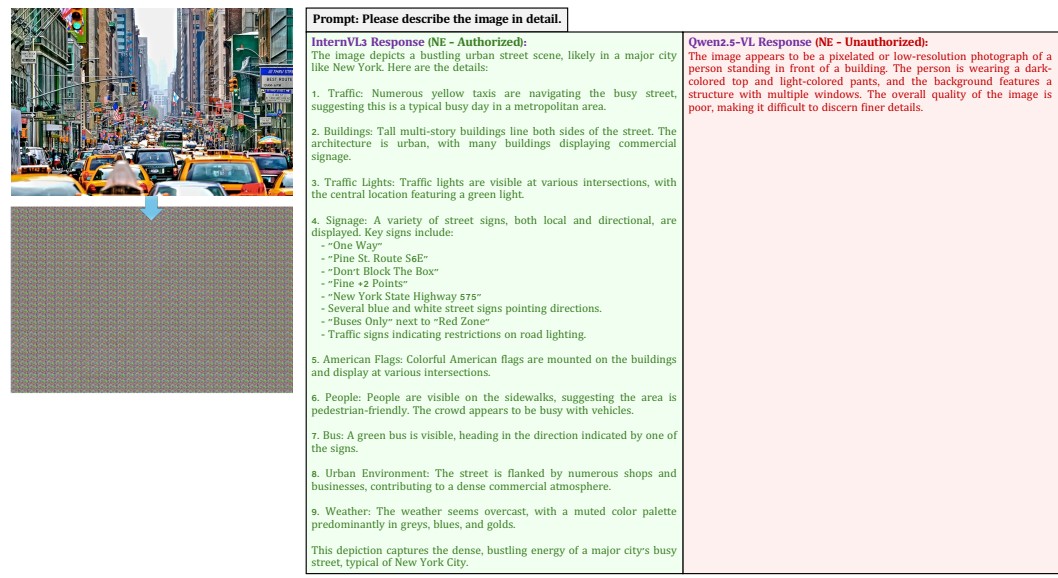

Figure 5: Illustrative visualization of effective on data authorization on VLM.

*partially* cancel the recoding and raise $m(f', \tilde{x}')$. This is realistic: linear projection does not require access to $f^\star$, only an estimate of the basis. However, recovery is imperfect in practice because acquisition and preprocessing (resize, compression, normalization, tokenization) generally do not commute with a fixed projector, and the authorized benignity relies on $f^\star$'s internal representations rather than pure input-space orthogonality. Thus, projection-back can reduce effect size but does not guarantee full restoration on arbitrary $f'$.

A stronger adversary with training-time control can *counter-adapt* by regularizing sensitivity (*e.g.,* encouraging larger singular components or Jacobian norms along estimated bases) so that the insensitivity directions shrink after training. This can realistically recover non-target utility, especially when the task admits redundancy. The trade-off is empirical rather than guaranteed: making all directions sensitive tends to increase brittleness and harms calibration/robustness on common corruptions, but such side effects may be acceptable to an attacker optimizing only $m(f', \tilde{x}')$. Additional practical limits include dependence on a modest probe budget for estimating the spectral basis; attenuation under aggressive acquisition pipelines (heavy crops or compression) or domain shift that changes early-layer geometry; and detectability, supervised or self-supervised detectors can learn to flag or strip structured, low-energy recodings when the basis is static. Finally, our analysis focuses on early layers and linearized views; extending guarantees to deeper Jacobians and temporally coupled modalities (audio/video) remains open.

