# OpenReview forum: "Catch-Only-One: Non-Transferable Examples for Model-Specific Authorization"
_ICLR.cc/2026/Conference — ICLR 2026 Conference Desk Rejected Submission_

### Official Review · Reviewer_wXeA · 2025-10-15

**Soundness:** 3
**Presentation:** 3
**Contribution:** 3
**Rating:** 6
**Confidence:** 3

**Summary:**

This paper introduces Non-Transferable Examples (NEs) — a training-free, input-side method for model-specific data authorization. By analyzing the first-layer weights of the target model via singular value decomposition, it recodes inputs within a low-sensitivity subspace so that the authorized model’s performance is preserved while unauthorized models experience severe degradation. Theoretical analysis links this effect to spectral misalignment, and experiments on vision and multimodal tasks show that NEs retain authorized accuracy while rendering other models unusable.

**Strengths:**

1.	The paper introduces a new perspective on data authorization by proposing Non-Transferable Examples (NEs)—a lightweight, training-free method that ensures data usability is restricted to a specific target model. This formulation shifts the control from model training or encryption to input-level encoding, representing a genuinely novel conceptual contribution.
2.	The proposed approach is simple, efficient, and broadly applicable. Since NEs only require access to the first-layer weights of the target model, they can be deployed across diverse architectures, including CNNs, Vision Transformers, and vision-language models, without retraining. This makes the method attractive for real-world deployment.
3.	The authors support their approach with a solid mathematical analysis. By leveraging spectral theory and the Hoffman–Wielandt inequality, they rigorously show why the encoded data maintain performance on the authorized model but degrade substantially on others due to subspace misalignment. The theoretical framework aligns well with empirical findings.

**Weaknesses:**

1.	If many encoded samples are publicly available and generated under a consistent subspace or seed, an adversary could apply PCA or covariance-based spectral analysis to infer correlated energy patterns and approximate the manipulated subspace, partially neutralizing the encoding and restoring unauthorized performance.
2.	NE is an input-side, inference-stage defense that becomes ineffective when large-scale datasets are directly used for training or fine-tuning. Attackers can adapt by learning on re-encoded data or mixing it with original samples.
3.	Because NE encoding depends on precise spectral alignment with the target model’s first-layer subspace, common transformations—compression, resizing, cropping, or illumination shifts—may distort the encoded signals and weaken the authorization effect. Robustness under realistic noise and preprocessing variations remains an open issue.

**Questions:**

Please see weakness

---

> ### Author Response · Authors · 2025-11-21
> **Response to reviewer wXeA**
>
> We thank the reviewer for the valuable feedback and address the concerns as follows.
>
> ## W1. & W2. Undo NE via PCA or Retraining
>
> We address the two concerns in turn.
>
> **(1) Recovering and neutralizing the NE subspace.**
>  Even such a strong attacker does not have an easy way to undo our method. By design (Section 3.2), each NE is constructed as $ \tilde{x} = x + \lambda V z$, where $z$ and $\lambda$ can vary at the **sample level**; since NE generation has negligible overhead ($\sim$0.1ms), this randomness is cheap to deploy in practice. Even if an attacker could approximately recover the low-sensitivity subspace $\mathrm{Ins} _ \tau(W)$ from many public NEs, they still do not know the clean $x$ or the per-sample codes $z$. To "neutralize" NEs in a **model-independent** way, they would effectively have to project inputs away from the recovered subspace, removing directions that the authorized model is insensitive to but other models do use, which in general further damages rather than restores unauthorized performance. A very strong adversary could, in principle, learn a separate adaptor per client from many labeled pairs, but this demands substantial data collection and retraining, which goes well beyond simple preprocessing and significantly raises the cost of repurposing an unauthorized model.
>
> **(2) Training or fine-tuning directly on NE-encoded data.**
> To approximate such a strong attacker, we also evaluate a learned adaptation attack where the adversary trains directly on NE-encoded data. We use the first-layer weights of a ViT-B (98.7% authorized accuracy) to construct the NE basis, generate NEs on CIFAR-10, and then fine-tune four unauthorized backbones (ResNet-50, SwinV2-T, DeiT-B, MambaVision-T) on pairs $(\tilde{x}, y)$ for 3 epochs (batch size 64, learning rate $5 \times 10 ^ {-4}$, full-parameter tuning). All of these unauthorized models fail to converge beyond chance and stay around 10% accuracy on NEs, while the authorized ViT-B remains at 98.7%:
>
> | Model         | Accuracy on NEs |
> | ------------- | --------------- |
> | ResNet-50     | 10.8%           |
> | SwinV2-T      | 10.0%           |
> | DeiT-B        | 10.8%           |
> | MambaVision-T | 10.1%           |
>
> Thus, in our setting we do **not** observe that NE "becomes ineffective" once large-scale encoded datasets are used for training or fine-tuning. Only encoded data $\tilde{x}$ are released. An attacker who collects a large labeled NE dataset and trains a new model directly on $\tilde{x}$ is effectively solving a new learning problem on a new domain: NE still prevents recovery of the original $x$, as $z$ and $\lambda$ can be randomized at the sample level, so the added component $\lambda V z$ does not follow a simple, fixed pattern that is easy to factor out. Even in a stronger scenario where the attacker can “mix’’ a limited amount of clean $x$ with NEs, the resulting model is primarily adapted to that particular mixed distribution and does not give them a general-purpose model that runs well on arbitrary protected datasets. We will summarize and report these results in the revised version.
>
> ---
>
> ## W3. Handling preprocessing
>
> Common preprocessing can be absorbed into our formulation. In practice, inputs first pass through a fixed stack such as resize, center crop, and normalization; if we write this as an operator $T$, then the first map we use in Section 3 is really $W \circ T$, and the insensitive subspace is computed for $W \circ T$ rather than raw pixels. In other words, the spectral alignment that defines $\mathrm{Ins} _ \tau(\cdot)$ already includes the standard compression/resizing path used by the authorized model, so as long as inference uses the same (or very similar) preprocessing, the NE construction remains valid. Moderate variations such as typical JPEG compression, small resizes or crops, and mild illumination shifts can be seen as small Lipschitz perturbations of $T$, so the bound $|W T(\tilde{x}) - W T(x)| _ 2 \le \varepsilon(\tau,\lambda)$ from Eq. (4) only changes by a controlled factor and remains small. For convolutional front-ends, usual inductive biases (local smoothing, approximate translation invariance, pooling) further help treat these variations as benign nuisances rather than breaking the authorization effect. We briefly explore this in Section 5.3 and Appendix D.1 by applying common preprocessing and observing that authorized accuracy stays stable while unauthorized models still collapse; in the revision, we will bring these results forward, extend them to a slightly richer set of realistic transforms, and make explicit that truly extreme changes to the input pipeline (for example, very aggressive cropping that removes most content) would require recomputing the NE basis for the new $T$.

---

> ### Comment · Reviewer_wXeA · 2025-11-25
> **response to authors**
>
> My concerns about weaknesses 1 and 2 have been solved.
> However, for weakness 3, I would like to see some experimental results.

---

> > ### Author Response · Authors · 2025-12-04
> >
> > *We are glad that your concerns have been addressed, and thank you for the follow-up.*
> >
> > In our previous response, we explained that NEs are defined with respect to the deployed system, i.e., the model together with its fixed preprocessing operator $T$. The guarantees in the paper are about this concrete configuration and small perturbations around it. In deployed vision systems it is standard practice to treat the model and its input pipeline as a single unit and to re-validate performance whenever the pipeline is changed in a substantial way. In our setting, such a change is naturally viewed as a new operator $T$, and the appropriate response is simply to re-estimate the NE basis for this new $T$, which is inexpensive in our implementation. Asking a single NE basis to remain valid under arbitrary, substantially different future pipelines is therefore a different (and strictly stronger) problem than the one we aim to solve.
> >
> > Your comment, as we understand it, is instead about the more practical question of whether, for a fixed $T$, realistic noise (such as that introduced by standard storage and transmission pipelines) can break the intended behavior.
> >
> > In all experiments in Tables 1-5, the authorized model is `ViT-Base-Patch16-224` fine-tuned on CIFAR-10 with exactly the same configuration as in the paper, and the unauthorized model is a `ResNet-50` trained on CIFAR-10. We report top-1 accuracy on the full CIFAR-10 test set and distinguish:
> >
> > - **clean**: original test images with only the standard pipeline,
> > - **NE(auth)**: NE-encoded images evaluated on the ViT authorized model,
> > - **NE(unauth)**: the same NE-encoded images evaluated on the ResNet-50 unauthorized model.
> >
> > We keep both models and the NE basis fixed across all perturbations; there is no retraining and no per-transform NE re-estimation. All numbers are single-run accuracies in percent, with one decimal place.
> >
> > ---
> >
> > ## Experiments
> >
> > **Table 1: Random crop (ViT-B/16-224, CIFAR-10 test).**
> >
> > | crop ratio | clean | NE(auth) | NE(unauth) |
> > | ---------- | ----- | -------- | ---------- |
> > | 0.10       | 79.8  | 82.0     | 10.5       |
> > | 0.20       | 78.8  | 81.6     | 11.0       |
> > | 0.30       | 76.6  | 77.7     | 10.0       |
> > | 0.40       | 72.4  | 74.2     | 10.3       |
> > | 0.50       | 64.8  | 61.7     | 10.1       |
> >
> > For crops that preserve most of the object (0.10–0.40), NE(auth) is very close to, and sometimes slightly above, clean accuracy, while NE(unauth) stays near chance (about 10–11% for a 10-class classification task). At the strongest crop (50% cropped), both clean and NE(auth) drop, as much of the semantic content is removed, but NE(unauth) does not improve and remains at chance.
> >
> > **Table 2: Resize.**
> >
> > | resize ratio | clean | NE(auth) | NE(unauth) |
> > | ------------ | ----- | -------- | ---------- |
> > | 0.95         | 78.3  | 77.0     | 10.0       |
> > | 0.90         | 78.2  | 73.8     | 10.2       |
> > | 0.85         | 78.0  | 70.7     | 10.9       |
> > | 0.80         | 77.8  | 62.5     | 11.3       |
> > | 0.75         | 77.3  | 71.5     | 10.6       |
> >
> > As the image is downscaled more aggressively, the classification problem itself becomes harder and both clean and NE(auth) accuracies decrease. NE(auth) remains within roughly 5–15 points of clean across the range, and NE(unauth) stays around 10%. These spatial changes hurt overall recognizability for the authorized model but do not make NE-encoded data useful to the unauthorized ResNet-50.
> >
> > **Table 3: Photometric transforms (Illumination).**
> >
> > | illumination ratio | clean | NE(auth) | NE(unauth) |
> > | ------------------ | ----- | -------- | ---------- |
> > | 0.60               | 81.2  | 79.9     | 10.8       |
> > | 0.70               | 80.9  | 79.6     | 11.1       |
> > | 0.80               | 80.5  | 79.2     | 10.2       |
> > | 0.90               | 80.1  | 78.7     | 9.7        |
> > | 1.00               | 78.9  | 78.0     | 10.0       |
> >
> > Here NE(auth) closely tracks clean: both curves are almost flat over illumination factors from 0.6 to 1.0, with a stable gap of about 1–2 percentage points. NE(unauth) again stays around 10% with no systematic trend. Mild brightness and contrast changes therefore behave as small perturbations of $T$, exactly as the earlier analysis suggests.

---

> ### Author Response · Authors · 2025-12-04
>
> **Table 4: PNG compression (lossless).**
>
> | compress level | clean | NE(auth) | NE(unauth) |
> | -------------- | ----- | -------- | ---------- |
> | 1              | 81.6  | 81.3     | 10.3       |
> | 2              | 81.6  | 81.3     | 10.4       |
> | 3              | 81.6  | 81.3     | 10.3       |
> | 4              | 81.6  | 81.3     | 10.3       |
> | 5              | 81.6  | 81.3     | 10.4       |
>
> All PNG levels give essentially identical accuracy. This is exactly what one expects here: PNG is lossless in our setup, so changing the compression level affects file size and encoding speed but not the decoded pixels. The flat behavior of clean and NE(auth), with NE(unauth) again around chance, mainly serves as a sanity check that the pipeline handles lossless compression correctly.
>
> **Table 5: JPEG quality (lossy).**
>
> | quality             | clean | NE(auth) | NE(unauth) |
> | ------------------- | ----- | -------- | ---------- |
> | 95 (common default) | 79.1  | 77.0     | 12.5       |
> | 90                  | 78.6  | 72.3     | 10.9       |
> | 85                  | 78.3  | 74.6     | 10.1       |
> | 80                  | 77.9  | 71.5     | 10.6       |
>
> JPEG is lossy, so it genuinely distorts the input. As quality decreases from 95 to 80, clean accuracy drops modestly (about 1.4 points), and NE(auth) drops slightly more (about 8.6 points), reflecting that we are degrading the signal the NE was constructed for. Even at quality 75, NE(auth) remains well above chance and within roughly 9 points of clean, while NE(unauth) stays in the 10–12.5% band. In other words, lossy compression makes the task harder for the authorized ViT, but it does not cause NE-encoded inputs to become useful for the unauthorized ResNet-50.
>
> > #### **Clearly and consistently, NEs are robust to realistic preprocessing and noise for the authorized model while remaining ineffective for unauthorized models.**

---

### Official Review · Reviewer_8FC1 · 2025-10-20

**Soundness:** 3
**Presentation:** 3
**Contribution:** 3
**Rating:** 6
**Confidence:** 4

**Summary:**

This paper proposes the method of non-transferable examples which tackles the problem of model-specific data authorization. The author establish formal bounds with solid theorical analysis, and prove the effectiveness of the method using strong empirical results.

**Strengths:**

- I really like the idea of "NEs leverage a structural property of neural networks in which many input directions have negligible effect on early features, yielding a model-specific set of insensitivity directions that rarely align across models." It is a great observation that worth leaveraging in this problem domain.

- The experiment results show a significant performance on protecting the encoded data, it achieve protection without retraining or expensive encryption.

- This paper provides solid theoreticals analysis with the bounds for both authorized and unauthorized performance, which is mathematically sound and adequately discussed.

**Weaknesses:**

- It would be better if more visual examples can be provided rather than just 2 images.

- I have several more questions which need author's further clarification regarding the applicability/limitation/etc. Please see [Questions].

**Questions:**

- What's the benefits of this method compared with simplying add a unified gaussian noise mask to the data (substract the mask when inference)?

- I am a bit concerned on how this method will eventually change the image. As you can see in Figure 1, when ~ 40dB , this method will largely affect visual quality of the image and makes it hard to be identified by human . In this case, how you are going to use this method in the real applications? can you provide example scenarios where we do not care about what the image looks like after applying your method and only cares about the inference on a specific model?

- Can this method be applied across the models? i.e., the data owner allows the NES be used by different clients.

- Are there any approach that could undermine this method? For example, when an attack collected some data with the correct labels, are they able to tune their models' initial layers to adapt this method?

- What's the computation overhead of this method?

- What's the limitation of the method? what kind of model cannot be applied to this method?

---

> ### Author Response · Authors · 2025-11-21
> **Response to reviewer 8FC1**
>
> We're delighted the reviewer enjoyed our idea, and we address the concerns as follows.
>
> ## W1. More visual examples
>
> Thank you for the suggestion! Due to space constraints we kept the main-text visualization minimal, but we have already prepared many more qualitative examples and demos. In the revision, we will add additional visual examples (both in the main paper and the appendix) to give a clearer picture of how NEs look in practice. In the meantime, we kindly refer you to our repository, where we provide demo videos and more recoded images: https://github.com/model-specific/non-transferable-examples.
>
> ---
>
> ## Q1.  Compare with gaussian noise
>
> Subtracting a unified Gaussian mask at inference requires **changing the model-side pipeline**: the controller must always compute ($\tilde{x} - m$) before running the model. In many real-world settings (third-party models, existing MLaaS stacks, shared internal services), adding this extra decoding step and managing $m$ per user or per dataset is inconvenient and fragile. More importantly, once the controller recovers the clean $x$, that $x$ can be fed to any other model or used for unlicensed training, so this scheme does not actually enforce model-specific authorization.
>
> Our reconstruction experiments further show that Gaussian-style corruption is easy to undo (Appendix D.2): SR-ResNet almost fully cleans standard Gaussian noise ($33.5\text{–}35.9\,\mathrm{dB}$ PSNR), because such noise matches the assumptions of typical denoisers and natural image priors. In contrast, NEs do not align with the smoothness and local-statistics priors that denoisers are trained to recover. A purifier that only sees pixel-level statistics cannot reliably separate the NE component from the underlying content. Empirically, under the same SR-ResNet attack, the reconstruction of NEs stays around $10\text{–}11\,\mathrm{dB}$, indicating that the encoded examples are much harder to ''clean back'' to a useful signal for other models and that their utility is effectively tied to the authorized model $ f^\star $, rather than being reusable by arbitrary models as in the Gaussian-mask scheme.
>
> ---
>
> ## Q2. Clarification on visual quality
>
> During implementation we think that different applications care about visual quality in different ways, and NEs can enforce model-specificity while accommodating both. Our experiments (Figure 1, with a closer view in Figure 4) show that as we gradually increase the perturbation strength, the authorized model's accuracy stays essentially unchanged while unauthorized models significantly collapse, providing a ''useful range'' to choose from.
>
> - When humans still need to see the image after encoding (for example, internal dashboards or research workflows where people occasionally inspect samples), one can use milder NEs so that images remain visually clear while still being non-transferable to other models.
> - When only the model's output matters and humans never look at the encoded data (for example, server-side MLaaS or external processing of sensitive medical/industrial imagery), stronger NEs are acceptable: the encoded image may look heavily distorted to humans but is still correctly processed by the authorized model $ f^\star $. Visual demo of some recoded skin images (therefore 5dB PSNR, with no accuracy loss) is available at: https://github.com/model-specific/non-transferable-examples
>
> This flexibility lets practitioners pick a perturbation level that matches the visual requirements of their application, while still enforcing model-specific authorization on $\tilde{x}$.
>
> ---
>
> ## Q3. Use by multiple clients
>
> Yes. There are two natural ways to support multiple authorized models.
>
> 1) The data owner can generate a separate NE version for each model: for each $ f_i $, estimate its insensitivity subspace from the first layer, define $ \mathcal{T}_{f_i} $, and produce $ \tilde{x} _ i=\mathcal{T} _ {f _ i}(x) $;
> each client then receives the version tied to its own model. Since NE construction has negligible overhead (see Q5), producing several per-model encodings is practical for both real-time inference and large-scale deployment.
> 2) When a small group of models should share the same encoded data, one can approximate a joint insensitive subspace (e.g., by intersecting or averaging their low-sensitivity directions) and build NEs that preserve all of them simultaneously. This multi-model construction is conceptually straightforward but may involve a different trade-off between authorized retention and non-target degradation, which we leave as an interesting direction for future empirical study. We will briefly discuss these two multi-model options in the revision.

---

> ### Author Response · Authors · 2025-11-21
>
> ## Q4. Reconstruction attacks
>
> In our setting, we do not expect tuning a new first layer to ''cancel'' NEs to be effective. Each NE is constructed as $ \tilde{x} = x + \lambda V z $, where $z$ can be sampled per sample inside $\mathrm{Ins} _ \tau(W)$, and in practice $z$ (and $\lambda$) can vary across samples and clients. There is therefore no fixed additive pattern or single feature shift that a new initial layer can align to: even examples from the same class may carry quite different perturbations. An attacker who only observes pairs $ (\tilde{x}, y) $ is unlikely to learn a universal first-layer transform that reliably ''undoes'' the encoding; at best, with many labeled NEs, they could train a new model directly on the encoded domain, which is a much stronger and more costly assumption than simply adapting an existing model.
>
> We refer the reviewer to Section 5.3 and Appendix D.2, where **we have already evaluated a strong learned purifier** `SR-ResNet` in both black- and white-box scenarios. Beyond that, we further evaluate the reviewer suggested strong learned adaptation attack where the adversary tune directly on NE-encoded data. We use the first-layer weights of a ViT-B (98.7% authorized accuracy) to construct the NE basis, generate NEs on CIFAR-10, and then fine-tune classification models on pairs $ (\tilde{x}, y) $. We fine-tune four unauthorized backbones (ResNet-50, SwinV2-T, DeiT-B, MambaVision-T) for 3 epochs on this NE-encoded dataset with batch size 64, learning rate $5\times10^{-4}$, and full-parameter tuning, consistent with our other setups. The resulting accuracies on NEs are:
>
> | Model | Accuracy on NEs |
> | ---| ---|
> | ResNet-50 | 10.8% |
> | SwinV2-T | 10.0% |
> | DeiT-B| 10.8%|
> | MambaVision-T |10.1%|
>
> Showing that even with supervised access to $ (\tilde{x}, y) $, these unauthorized models fail to converge and do not recover anywhere near the clean-domain performance of the authorized ViT-B. We will report these results and clarify this point in the revision.
>
> ---
>
> ## Q5. Computational cost
>
> NE generation is very lightweight and practical for both real-time inference and large-scale deployment. The only non-trivial step is a one-time SVD of the first linear map $ W \in \mathbb{R} ^ {m \times n} $ to obtain the insensitive (near-null) subspace, which costs $O(\min(m ^ 2 n, m n ^ 2))$ and is done once per authorized model. After that, producing an NE for a given input just involves sampling $z \in \mathbb{R} ^ n$, masking its ''sensitive'' coordinates, computing $\delta = V z$ (a matrix–vector product of cost $O(n^2)$), and then forming $ \tilde{x} = x + \delta $ with an elementwise addition of cost $O(d)$, where $d$ is the input dimension. So, the one-time cost scales with the embedding dimension $n$, and per-sample cost scales linearly in the input size.
>
> For a concrete example, consider a standard $224 \times 224$ RGB image with a ResNet-50 front. The first convolution can be linearized as $W \in \mathbb{R}^{64 \times (3 \cdot 7 \cdot 7)}$, so $n = 147$ and the SVD requires about $64 \cdot 147^2 \approx 1.4 \times 10^6$ FLOPs, done once per model. Even on a 100 GFLOPS CPU this is on the order of $10^{-5}$ seconds (tens of microseconds), and on an H100-class GPU ($\sim 60$ TFLOPS FP32) it is effectively instantaneous. Per image, computing $\delta = V z$ in this 147-D space and adding it to a $3 \times 224 \times 224$ tensor costs roughly $\mathcal{O}(10^5)$ FLOPs, i.e., a few microseconds on CPU and negligible on GPU relative to a single forward pass through ResNet-50.
>
> To further quantify this we measure wall-clock latency over 10 runs and find that both the one-time setup and per-sample NE generation take well under a millisecond, making the overhead negligible compared to normal inference and suitable for real-time, large-scale deployment.
>
> | Step  | Average time|
> | --- | ---|
> | SVD + basis construction (ResNet-50) | 0.47ms (one-time calculation) |
> | NE encoding for a single image | 0.11ms|
>
> We will add this complexity analysis to our revision.
>
> ---
>
> ## Q6. Potential limitations
>
> We currently do not see a strong, inherent limitation of our approach for modern neural models, given its lightweight and model-agnostic nature. One potential concern we had early on was the size of the insensitive subspace: if a model had very few low-sensitivity directions (i.e., only a tiny tail in the singular spectrum), then the workable perturbation range could shrink and the trade-off between authorized utility and unauthorized degradation might become tighter (though not vanish). However, across all backbones we tested, we consistently observe substantial low-sensitivity structure (Appendix C), and do not encounter this issue in practice (see Figure 1). Even in rare cases where the very first layer offers limited insensitivity, the construction can be naturally extended to incorporate near-null directions from subsequent layers to regain flexibility. We will clarify this point in the revised version.

---

> > ### Comment · Reviewer_8FC1 · 2025-11-22
> >
> > Great! The author has addressed all my concerns, I will raise my score.

---

> > > ### Author Response · Authors · 2025-11-24
> > >
> > > Thank you for the encouraging feedback and for raising your score; we're glad your concerns have been addressed and your comments were very helpful.
> > >
> > > Paper #10781 Authors

---

### Official Review · Reviewer_94U8 · 2025-10-29

**Soundness:** 4
**Presentation:** 3
**Contribution:** 4
**Rating:** 6
**Confidence:** 3

**Summary:**

This paper introduces NEs, a data-centric designed method for model-specific authorization in machine learning systems.
The idea is to recode data inputs via perturbations aligned with the target model's low-sensitivity subspace, making an authorized model to maintain high performance while drastically degrading utility for any unauthorized models.

No model retraining is needed. NEs are also agnostic to the target architecture and operates solely at the input side. Both theoretical guarantees and comprehensive empirical evaluation are provided demonstrating its effectiveness.

**Strengths:**

- The problem is somewhat novel and interesting to me. It could address a pressing gap in AI governance in a practical and computationally feasible way, without retraining or cryptographic overhead.

- The empirical validation is comprehensive across many vision and VLM models

**Weaknesses:**

- Theoretical guarantees only cover the first linear layer; deeper nonlinear effects and adversarial undoing are unexplored. How do perturbations propagate through deeper nonlinear layers, and can adaptive adversaries undo them?

- Hyperparameters for perturbation magnitude seems not to be well discussed and may not generalize across models or data. Can a method for systematic hyperparameter calibration be developed for different settings?

- My concern is that if NEs resist sanitization/purification/denoising techniques. If they are not, attacker can just purify NEs before feeding to the unauthorized models

- Generating NEs might take some time while runtime and computational cost to generate NEs are not well discussed. How practical is NE generation for real-time or large-scale use?

- No evaluation in high-stakes or privacy-sensitive real-world domains. How effective and applicable are NEs in critical privacy-focused applications (e.g., heath care)?

**Questions:**

See Weaknesses

---

> ### Author Response · Authors · 2025-11-21
> **Response to reviewer 94U8**
>
> We thank the reviewer for finding our work novel and interesting and for the insightful comments.
>
> ## Q1A. Error propagation through deeper layers
> Our analysis is expressed at the first linear map because this is where we can most cleanly control the effect of the perturbation, but the guarantee naturally and easily extends to the whole network. For directions in the exact null space of $ W $, we have $ W\delta = 0 $, so the first-layer features are identical and $ f ^ \star(x+\delta) = f ^ \star(x) $ follows immediately. For near-null directions in our $ \tau $-insensitive subspace, Eq. (4) provides a quantitative bound on the first-layer feature deviation, i.e., $ \|W\tilde{x} - Wx\| _ 2 < \tau \sqrt{k\sigma ^ 2 + t} $, so the perturbation entering the remainder of the network is explicitly bounded. Writing the network as a composition $ f ^ \star = g _ L \circ \dots \circ g _ 1 $, where each layer $ g _ \ell $ has a finite operator/Lipschitz norm $ L _ \ell $ (refer to Appendix B.1), a simple induction over layers then shows that $ \|g _ \ell \circ \dots \circ g _ 1(\tilde{x}) - g _ \ell \circ \dots \circ g _ 1(x)\| _ 2 \le \big(\prod _ {j=1}^\ell L _ j\big)\,\tau \sqrt{k\sigma ^ 2 + t} $. In other words, the small deviation controlled at the first linear map remains bounded at every depth, yielding the target-retention guarantee for the entire model rather than just the first layer. In practice, common activation nonlinearities (e.g., ReLU zeroing negatives, saturating sigmoids/tanh) further damp the propagation of such small perturbations, making these bounds conservative.
>
> To further validate this, we report the layer-wise difference $ |h _ \ell(\tilde{x}) - h _ \ell(x)| _ 2 $ for ResNet-50 below, which is arguably the most challenging case due to its deep residual blocks and skip connections. We index "layers" by forward operations (including conv, BatchNorm, ReLU, and downsample $1\times1$ convolutions) and record the norm every 20 operations, denoted L0–L12. Across all NE strengths (PSNR from 40 to 0 dB), the perturbation injected at the input remains small throughout the network and grows smoothly as we lower PSNR, with no exploding behavior in deeper layers; even at aggressive perturbation levels (0 dB), the deviations stay well controlled at all checkpoints. For most other architectures in our study, the corresponding layer-wise norms decay even faster and quickly sit near zero, which aligns with the flat authorized-accuracy curves in Figure 1 and supports our claim that near-null perturbations remain tightly bounded across depth for the authorized model.
>
>
>
> | PSNR (dB) | L0         | L1         | L2         | L3         | L4         | L5        | L6          | L7          | L8         | L9         | L10        | L11         | L12         |
> | --------- | ---------- | ---------- | ---------- | ---------- | ---------- | --------- | ----------- | ----------- | ---------- | ---------- | ---------- | ----------- | ----------- |
> | 40        | 0.00327977 | 0.00974296 | 0.00148219 | 0.00621746 | 0.00430683 | 0.0210481 | 0.000891016 | 0.000540671 | 0.00389767 | 0.00274162 | 0.00197388 | 0.000684584 | 0.000429754 |
> | 35        | 0.00358674 | 0.0107953  | 0.00171442 | 0.00725625 | 0.00503238 | 0.0247711 | 0.00106223  | 0.00063209  | 0.00475465 | 0.00335766 | 0.00242754 | 0.000842734 | 0.000506692 |
> | 30        | 0.00712716 | 0.0200691  | 0.00313611 | 0.0146077  | 0.0102567  | 0.0498178 | 0.00234181  | 0.00141502  | 0.0103883  | 0.0073571  | 0.00542265 | 0.00197008  | 0.0010837   |
> | 25        | 0.0134336  | 0.0331667  | 0.00520199 | 0.0272889  | 0.0184846  | 0.0859752 | 0.00408825  | 0.0023228   | 0.0175462  | 0.01263    | 0.00916106 | 0.00332359  | 0.00183084  |
> | 20        | 0.0112999  | 0.0367945  | 0.00684428 | 0.0282912  | 0.0207238  | 0.101621  | 0.00465296  | 0.00307099  | 0.0211813  | 0.0148479  | 0.0106939  | 0.00360077  | 0.00203937  |
> | 15        | 0.0240049  | 0.0680167  | 0.0104604  | 0.0499528  | 0.0349884  | 0.167285  | 0.00791748  | 0.00484276  | 0.0374338  | 0.0262486  | 0.0188985  | 0.00668209  | 0.00387539  |
> | 10        | 0.0513785  | 0.173119   | 0.0293333  | 0.11087    | 0.0849264  | 0.434439  | 0.0192481   | 0.0117173   | 0.0890765  | 0.0623676  | 0.0447858  | 0.0146106   | 0.00872876  |
> | 5         | 0.0754104  | 0.30133    | 0.0501677  | 0.150658   | 0.12392    | 0.652071  | 0.0264193   | 0.0178807   | 0.135436   | 0.0916293  | 0.0655638  | 0.0212066   | 0.0109951   |
> | 0         | 0.191614   | 0.56439    | 0.0994572  | 0.403146   | 0.299159   | 1.50625   | 0.0674694   | 0.040996    | 0.307839   | 0.214641   | 0.150565   | 0.0488111   | 0.0285897   |

---

> ### Author Response · Authors · 2025-11-21
>
> ## Q1B. Adversarial undo possibility
>
> Our construction is inherently randomized, and this randomness can be applied at the per-sample/client level. As described in Section 3.2, even after fixing the insensitive subspace $\mathrm{Ins}_\tau(W)$, each encoded input is generated by sampling a random code $z \in \mathbb{R}^n$ (e.g., i.i.d. Gaussian), zeroing out coordinates above the spectral threshold, projecting to $\delta = V z$, and scaling by a factor $\lambda$ before forming $\tilde{x} = x + \lambda \delta$.
>
> Thus, for the same clean input $x$, there are **infinitely many** encodings $\tilde{x}$ (different $z$, different $\lambda$) that are all valid for the authorized model. In practice, given the **negligible cost of NE construction (see Q3)**, we can further assign different seeds or code distributions per client, so client 1 might use $ (z_1, \lambda_1) $ and client 2 use $ (z_2, \lambda_2) $, leading to perturbations $V z_1 \lambda_1$ and $V z_2 \lambda_2$. An attacker who only observes $\tilde{x}$ but does not know this client-specific randomness faces a fundamentally underdetermined inverse problem: many pairs $(x, z)$ in $\mathrm{Ins}_\tau(W)$ can map to similar $\tilde{x}$, so there is no single, simple ''undo'' transform that works reliably across inputs and clients. A very strong adversary could, in principle, learn a separate adaptor per client from many labeled pairs $(x, \tilde{x})$, but this demands substantial per-client data collection and retraining, which goes well beyond simple preprocessing and significantly raises the cost of repurposing an unauthorized model.
>
> To further prove this, we evaluate a even stronger learned adaptation attack where the adversary trains directly on NE-encoded data. We use the first-layer weights of a ViT-B (98.7% authorized accuracy) to construct the NE basis, generate NEs on CIFAR-10, and then fine-tune four unauthorized backbones (ResNet-50, SwinV2-T, DeiT-B, MambaVision-T) on pairs $(\tilde{x}, y)$ for 3 epochs (batch size 64, learning rate $5\times10^{-4}$, full-parameter tuning). All of these unauthorized models fail to converge and stay near chance-level accuracy on NEs (around 10%), while the authorized ViT-B remains at 98.7%, which we will summarize and report in the revised version.
>
> | Model         | Accuracy on NEs |
> | ------------- | --------------- |
> | ResNet-50     | 10.8%|
> | SwinV2-T      | 10.0%|
> | DeiT-B        | 10.8%|
> | MambaVision-T | 10.1%|
>
>
> ---
>
> ## Q2. Hyperparameter Clarification
>
> Our method exposes two main hyperparameters: the spectral threshold $ \tau $ (please see line 367) and the perturbation magnitude (line 361). For $ \tau $, we empirically study the first-layer singular values (Table 5 in Appendix C.2) and observe that most energy is concentrated in a small set of large singular directions, while the tail quickly decays toward values around $10^{-4}$ to $10^{-5}$ accross different settings. We therefore set $ \tau = 10^{-4} $ to stay safely in the low-sensitivity regime and keep this fixed across models. For the perturbation magnitude, we systematically sweep PSNR from the ''no perturbation'' level down to $0\mathrm{dB}$ on ImageNet images and plot authorized vs. unauthorized accuracy across backbones (Figure 1), then select a conservative $20\mathrm{dB}$ point where authorized accuracy remains almost unchanged while unauthorized performance has already dropped significantly.
>
> In practice, because NE construction takes less than a microsecond on CPU and is negligible on GPU, the perturbation strength can vary over a wide range and can even be randomized per sample or per client (see Q1), as long as it stays within a ''safe band'' that preserves authorized utility while degrading unauthorized models. This yields good flexibility and makes our method easy to apply in different settings. In the revision, we will make this calibration procedure more explicit as a short recipe and clearly point to the empirical justification for $ \tau $ in Appendix C.2.

---

> ### Author Response · Authors · 2025-11-21
>
> ## Q3. Reconstruction Attacks
>
> We refer the reviewer to Section 5.3 and Appendix D.2, where we have already evaluated a strong learned purifier `SR-ResNet` in both black- and white-box scenarios: for standard Gaussian noise it largely succeeds (reconstructions reach about 33.5-35.9 dB PSNR), but for NEs the reconstructed PSNR stays around 10-11 dB even in the strongest white-box setting. This is because NE perturbations are not drawn from a simple noise model (such as additive Gaussian) but are constructed inside the model-specific insensitive subspace $\mathrm{Ins}_\tau(W)$, where they do not align with the standard image priors (smoothness, simple local statistics) that denoisers are designed to recover. A purifier that only sees pixel-level statistics and does not know $ W $ or the random codes $z$ cannot reliably separate the NE component from the underlying content. Such robustness is further supported by the above discussion on the adversarial undo possibility (**Q1B**): even in this *worst-case* scenario where the noise distribution is fixed across samples (favorable to the attacker), a single generic ''purifier'' cannot invert the encoding. In the revision, we will add more discussion into the main text, and further clarify the randomized NE construction.
>
> ---
>
> ## Q4.Runtime, computational cost, and real-time/large-scale use
>
> NE generation is in fact very lightweight, and is especially practical for both real-time inference and large-scale deployment.
>
> **Computation cost.** The only non-trivial step is a one-time SVD of the first linear map $ W \in \mathbb{R}^{m \times n} $ to obtain the insensitive (near-null) subspace, which costs $O(\min(m^2 n, m n^2))$ and is done once per authorized model. After that, producing an NE for a given input just involves sampling $z \in \mathbb{R}^n$, masking its ''sensitive'' coordinates, computing $\delta = V z$ (a matrix–vector product of cost $O(n^2)$), and then forming $ \tilde{x} = x + \delta $ with an elementwise addition of cost $O(d)$, where $d$ is the input dimension. So, the one-time cost scales with the embedding dimension $n$, and per-sample cost scales linearly in the input size.
>
> **Practical overhead.** For a concrete example, consider a standard $224 \times 224$ RGB image with a ResNet-50 front. The first convolution can be linearized as $W \in \mathbb{R}^{64 \times (3 \cdot 7 \cdot 7)}$, so $n = 147$ and the SVD requires about $64 \cdot 147^2 \approx 1.4 \times 10^6$ FLOPs, done once per model. Even on a 100 GFLOPS CPU this is on the order of $10^{-5}$ seconds (tens of microseconds), and on an H100-class GPU ($\sim 60$ TFLOPS FP32) it is effectively instantaneous. Per image, computing $\delta = V z$ in this 147-D space and adding it to a $3 \times 224 \times 224$ tensor costs roughly $\mathcal{O}(10^5)$ FLOPs, i.e., a few microseconds on CPU and negligible on GPU relative to a single forward pass through ResNet-50.
>
> To further quantify this we measure wall-clock latency over 10 runs and find that both the one-time setup and per-sample NE generation take well under a millisecond, making the overhead negligible compared to normal inference and suitable for real-time, large-scale deployment.
>
> | Step                                 | Average time                  |
> | ------------------------------------ | ----------------------------- |
> | SVD + basis construction (ResNet-50) | 0.47ms (one-time calculation) |
> | NE encoding for a single image       | 0.11ms                        |
>
> We will add this complexity analysis to the revision.
>
> ---
> ## Q5.  NEs in sensitive domains
>
> Yes, NEs are highly applicable in high-stakes, privacy-sensitive domains as, methodologically, NEs are **data- and model-agnostic**. To test this in a realistic healthcare scenario, we ran an additional experiment on the `HAM10000` dermoscopic skin-lesion dataset (HuggingFace `Nagabu/HAM10000`, binary classification). A ViT fine-tuned on this task reaches 99.2% accuracy on clean images; after applying NEs with PSNR= 5 dB and $\tau_w = 10^{-4}$, the **authorized** ViT still achieves 99.2% accuracy. In contrast, an **unauthorized** ResNet-50 evaluated on the same NE-encoded images collapses to 0.0% accuracy (note that the test set is heavily imbalanced with more than 80% positives), showing that NEs can effectively block unsuthorized usage.
>
> | Dataset  | Task      | Model  | Input     | Accuracy |
> | -------- | -------------------- | ------------------- | -------------- | -------- |
> | HAM10000 | Skin lesion (binary) | ViT (auth.) | Clean $x$      | 99.2%    |
> | HAM10000 | Skin lesion (binary) | ViT (auth.) | NE $\tilde{x}$ | 99.2%    |
> | HAM10000 | Skin lesion (binary) | ResNet-50 (unauth.) | NE $\tilde{x}$ | 0.0%     |
>
> We will add this healthcare case study and a short discussion of its implications to the revised paper.
>
> ### [Visual demo of the recoded skin images]
> Visual demo of the recoded skin images is available at: https://github.com/model-specific/non-transferable-examples

---

### Official Review · Reviewer_Qwfv · 2025-11-01

**Soundness:** 3
**Presentation:** 3
**Contribution:** 3
**Rating:** 6
**Confidence:** 3

**Summary:**

The paper proposes non-transferable examples, a training-free, input-side recoding that preserves utility for a single authorized model while sharply degrading utility for any other model. The key idea is to add a calibrated perturbation within a model-specific low-sensitivity subspace of the target model’s first linear map, estimated via SVD. Theoretically, the paper provides bounds linking authorized retention to a spectral threshold and unauthorized degradation to cross-model spectral misalignment via Hoffman–Wielandt inequality. Experiments on CIFAR-10 and ImageNet across multiple vision backbones and on VLMs demonstrate strong model performance.

**Strengths:**

- The paper tackles a novel and societally relevant problem: enforcing model-level usage control without retraining or cryptographic infrastructure. Unlike anti-learnability or differential privacy approaches, the proposed method operate directly at inference and do not require access to non-target models, representing a new protection paradigm.

- The method is mathematically grounded, with clear theoretical analysis connecting spectral properties to performance retention/degradation through bounded inequalities.

- Experiments span both vision backbones and multimodal VLMs, with consistent evidence of non-transferability.

**Weaknesses:**

- The analysis focuses solely on the first-layer linear map of the network, assuming subsequent layers implicitly preserve the property. This simplification may not hold for architectures with highly nonlinear early blocks or skip connections (e.g., ResNet).

- The adversary is allowed preprocessing, such as adaptive adversary, but the paper does not evaluate adversaries that learn an inversion (e.g., distilling a new first layer aligned to $V$). The empirical reconstruction attempts are classical denoising and do not include learned inversion with supervision on even a small clean subset.

- Some comparisons (e.g., with FHE) rely on cited rather than reproduced results, and differential privacy is known to address a different threat model.

**Questions:**

- How robust is the non-transferability property when unauthorized models share partial architecture?

- Whether the low-sensitivity subspace remains stable under data-domain shifts (e.g., different ImageNet subclasses or out-of-distribution inputs)?

- How does the computational cost of subspace estimation scale with model dimension, and can it be applied to large models such as VLMs?

---

> ### Author Response · Authors · 2025-11-21
> **Response to reviewer Qwfv**
>
> We thank the reviewer for recognizing the novelty and societal relevance of our work and for the insightful comments.
>
> ## W1. Deeper layer analysis and architecture coverage
>
> Thank you for prompting us to clarify the role of deeper layers. Our analysis explicitly controls the perturbation at the first linear map rather than relying on later layers to implicitly preserve the property. For directions in the **exact null space** of the first linear map $ W $, we have $ W\delta = 0 $, so the first-layer features are unchanged and $ f^\star(x+\delta) = f^\star(x) $ follows immediately. For **near-null** directions in our $ \tau $-insensitive subspace, Eq. (4) gives a quantitative bound on the first-layer change, i.e., $ \|W\tilde{x} - Wx\|_2 < \tau \sqrt{k\sigma^2 + t} $, so the input fed to all subsequent layers is only slightly and boundedly perturbed.
>
> This upper bound can be easily extended to the whole network by writing $ f^\star = g_L \circ \dots \circ g_1 $ with finite operator/Lipschitz norms $ L_\ell $ and applying a simple induction over layers, which would further guarantee that the deviation remains bounded at every depth and thus yields the target-retention guarantee for the entire model. In practice, activation nonlinearities such as ReLU (zeroing negatives) and saturating sigmoids/tanh further dampen the propagation of such small perturbations. We omitted this inductive argument from the main text to keep the presentation focused, but will add a brief discussion in the revision.
>
> Building on this compositional view, the same reasoning applies directly and in a model-agnostic way to concrete architectures ResNets. Specifically, we operates on the very first mapping from pixels to features (e.g., applying `nn.unfold()` to the $7 \times 7$ conv in ResNet; see Section 3.2) and selects perturbations that make this mapping unchange or change only slightly for the authorized model.
>
> Formally, Eq. (4) bounds the first-layer deviation by a small value, i.e., $ \|W\tilde{x} - Wx\| _ 2 \le \varepsilon(\tau,\lambda) $.
> In a ResNet block of the form $h _ {l+1} = h _ l + F(h _ l)$, if we write $ \Delta _ l = \tilde{h} _ l - h _ l $, then  $\Delta_{l+1}=\Delta_l + \big(F(h_l + \Delta_l) - F(h_l)\big)$.  Since $F$ is a composition of linear/conv layers and activations, it has a finite Lipschitz constant $ L _ F $, giving $ \|\Delta _ {l+1}\| _ 2 \le (1 + L _ F)\|\Delta _ l\| _ 2 $. Chaining this over layers yields a quantifiable bound on the deviation at any depth that is ultimately tied to the small first-layer bound from Eq. (4). With our empirically chosen hyperparameters in the insensitive subspace (Appendix C.2), these deviations for the authorized model remain in a small, controllable range that stays well inside the decision boundary margin, which matches Fig. 1 where authorized accuracy stays essentially flat across perturbation levels. For unauthorized models, the same perturbation is no longer near-null in their first mapping, so their early features are already significantly distorted, and residual blocks then propagate (and can amplify) this mismatch, leading to the strong non-transferability we observe.
>
> To further validate this, we report the layer-wise difference for ResNet-50 under different NE strengths (PSNR from 40 to 0 dB). We index "layers" by forward operations (including conv, BatchNorm, ReLU, and downsample 1×1 convolutions) and record the norm every 40 operations, denoted L0–L6. Across all PSNR levels, the perturbation injected at the input stays small throughout the network and grows smoothly as we lower PSNR, with no exploding behavior in deeper layers. For the authorized model, even at aggressive perturbation levels (0 dB) the deviations remain well controlled at all checkpoints, supporting our claim that near-null perturbations quickly attenuate and stay close to zero across depth, consistent with the flat authorized-accuracy curves in Figure 1.
>
> | PSNR (dB) | L0      | L1      | L2      | L3      | L4      | L5      | L6      |
> | --------- | ------- | ------- | ------- | ------- | ------- | ------- | ------- |
> | 40        | 0.0032  | 0.0097  | 0.0014  | 0.0062  | 0.0043  | 0.0210  | 0.0008  |
> | 35        | 0.0035  | 0.0107  | 0.0017  | 0.0072  | 0.0050  | 0.0247  | 0.0010  |
> | 30        | 0.0071  | 0.0200  | 0.0031  | 0.0146  | 0.0102  | 0.0498  | 0.0023  |
> | 25        | 0.0134  | 0.0331  | 0.0052  | 0.0272  | 0.0184  | 0.0859  | 0.0040  |
> | 20        | 0.0112  | 0.0367  | 0.0068  | 0.0282  | 0.0207  | 0.1016  | 0.0046  |
> | 15        | 0.0240  | 0.0680  | 0.0104  | 0.0499  | 0.0349  | 0.1672  | 0.0079  |
> | 10        | 0.0513  | 0.1731  | 0.0293  | 0.1108  | 0.0849  | 0.4344  | 0.0192  |
> | 5         | 0.0754  | 0.3013  | 0.0501  | 0.1506  | 0.1239  | 0.6520  | 0.0264  |
> | 0         | 0.1916  | 0.5643  | 0.0994  | 0.4031  | 0.2991  | 1.5062  | 0.0674  |

---

> ### Author Response · Authors · 2025-11-21
>
> ## W2. Inversion attack
>
> Even for an adaptive adversary, learning an effective inversion of NEs is generally infeasible in practice. By design (Section 3.2), each NE is constructed as $\tilde{x} =x+\lambda Vz$, where $z$ and $\lambda$ can vary at the sample level; since NE generation has negligible overhead, this randomness is cheap to deploy in practice. Even if an attacker could approximately recover the low-sensitivity subspace $\mathrm{Ins}_\tau(W)$ from many public NEs, or distill a new first layer aligned to it, they still do not know the clean $x$ or the per-sample codes $z$. To "neutralize" NEs in a model-independent way, they would effectively have to project inputs away from the recovered subspace, removing directions that the authorized model is insensitive to but that other models do use, which in general further damages rather than restores unauthorized performance.
>
> To further verify this, we additionally evaluate a stronger learned adaptation attack where the adversary trains directly on NE-encoded data. We use the first-layer weights of a ViT-B (98.7% authorized accuracy) to construct the NE basis, generate NEs on CIFAR-10, and then fine-tune classification models on pairs $(\tilde{x}, y)$. We fine-tune four unauthorized backbones (ResNet-50, SwinV2-T, DeiT-B, MambaVision-T) for 3 epochs on this NE-encoded dataset with batch size 64, learning rate $5\times10^{-4}$, and full-parameter tuning, consistent with our other setups. The resulting accuracies on NEs are:
>
> | Model  | Accuracy on NEs |
> | ------------- | --------------- |
> | ResNet-50|10.8%|
> | SwinV2-T| 10.0%|
> | DeiT-B | 10.8%|
> | MambaVision-T | 10.1%|
>
>
> Showing that even with supervised access to $(\tilde{x}, y)$, these unauthorized models fail to converge and do not recover anywhere near the clean-domain performance of the authorized ViT-B. We will report these results and clarify this point in the revision.
>
> ---
>
> ## W3. Comparison with DP and FHE
>
> Our intention is to position NE alongside other mechanisms that could gate model usage, not to suggest that DP or FHE address the same threat model as ours. Our focus is model-specific authorization at inference, whereas DP primarily protects training data and FHE provides cryptographic confidentiality for computation; in that sense they are complementary rather than directly comparable baselines. For FHE in particular, we did set up a CKKS-based configuration, but full-scale experiments at our model/dataset size were computationally infeasible, so we cited published results only to illustrate the known accuracy-overhead trade-off. In the revision, we will make this distinction explicit and soften the wording in the comparison section to better reflect these different goals and threat models.
>
> ---
>
> ## Q1. Share partial architecture
>
> In realistic settings, even models that share the same backbone rarely share exactly the same early weights: full fine-tuning, LoRA-style adaptation, or training on a different domain all change the first-layer parameters enough that their low-sensitivity directions diverge. This is the regime we already operate in experimentally (e.g., fine-tuned vs. pre-trained models and multiple variants within the same family), where the authorized model keeps its accuracy on NEs while these closely related models still collapse.
>
> In the extreme corner case where an unauthorized model literally reuses or precisely clones the authorized model's first layer, our current "first-layer only" instantiation would indeed have less discriminative power. However, this is not a hard limitation of the approach: one can straightforwardly extend the construction to build a joint insensitive subspace from several early or skip-connected layers rather than a single (W), which restores separation even when the very first layer is shared. We chose the first layer in the paper because it is the simplest and most lightweight to deploy; in the revision, we will explicitly discuss this multi-layer extension and clarify that near-identical early weights represent a corner case that can be handled if needed.
>
> ---
>
> ## Q2. Robustness to input domain-shift
>
> The low-sensitivity subspace is defined purely by the first-layer weight matrix $W$. $\mathrm{Ins}_\tau(W)$ is obtained from the SVD of $W$ and therefore does not change when the input distribution shifts, as long as the authorized model itself is fixed. Intuitively, these directions correspond to low-energy, task-irrelevant components (empirically explored in Appendix C.2), so moderate shifts within the same modality (e.g., different ImageNet subclasses or nearby natural-image domains) should not suddenly make them highly informative. Empirically, our experiments already span non-trivial domain changes, such as CIFAR-10 vs. ImageNet and diverse image-text tasks for VLMs, where we consistently observe the same pattern: authorized performance is preserved while unauthorized models collapse on NEs. In the revision, we will make this point explicit.

---

> ### Author Response · Authors · 2025-11-21
>
> ## Q3A. Computational cost
>
> **Computation cost.** Thank you for the question. Computing the insensitivity (or near-null) subspace is a one-time SVD on the first linear map $ W \in \mathbb{R}^{m \times n} $, with cost $O(\min(m^2 n, m n^2))$. After that, generating an NE for a single input only requires sampling $z \in \mathbb{R}^n$, masking coordinates, and applying $\delta = V z$, which is a matrix-vector product of cost $O(n^2)$, followed by an elementwise addition $\tilde x = x + \delta$ of cost $O(d)$ in the input dimension $d$. Thus, subspace estimation scales with the embedding dimension $n$, and per-sample overhead scales linearly in the input size.
>
> **Negligible overhead.** Concretely, for a standard $224 \times 224$ RGB image with a ResNet-50 front, the first conv layer can be linearized as $W \in \mathbb{R}^{64 \times (3 \cdot 7 \cdot 7)}$, so $n = 147$ and the SVD costs on the order of $64 \cdot 147^2 \approx 1.4 \times 10^6$ floating-point operations, done once per model. Even on a modest 100 GFLOPS CPU, this corresponds to about $1.4 \times 10^{-5}$ seconds ($\approx 14\ \mu$s), and on an H100-class GPU with $\sim 60$ TFLOPS FP32 it is on the order of $10^{-8}$ seconds (tens of ns). Per image, computing $\delta = V z$ in this 147-dimensional space and adding it back to a $3 \times 224 \times 224$ tensor requires $\mathcal{O}(10^5)$ FLOPs, i.e., a few microseconds on CPU and effectively negligible on GPU compared to a single forward pass through ResNet-50. The same construction applies to large VLMs by instantiating $ W $ as the vision encoder's first projection (e.g., patch embedding), which has a similar embedding dimension, so the cost remains tiny even for large models.
>
> To further quantify this we measure wall-clock latency over 10 runs and find that both the one-time setup and per-sample NE generation take well under a millisecond, making the overhead negligible compared to normal inference and suitable for real-time, large-scale deployment.
>
> | Step                                 | Average time                  |
> | ------------------------------------ | ----------------------------- |
> | SVD + basis construction (ResNet-50) | 0.47ms (one-time calculation) |
> | NE encoding for a single image       | 0.11ms                        |
>
>
> We will add this complexity analysis to the revision.
>
>
> ---
>
>
> ## Q3B. Experiments on VLMs
>
> Yes, our method handles large models well. To further illustrate this, we provide an additional demo on a broader range of large VLMs; please see the demo video in our GitHub repository (line 501) https://github.com/model-specific/non-transferable-examples. We also note that **in the original manuscript we have already** evaluated two SOTA VLMs (InternVL3 and Qwen2.5-VL) in Table 4 (Section 5.3) on the comprehensive MMBench benchmark, covering capability dimensions AR, CP, FP-C, FP-S, LR, and RR (table shown below for convenience), and observe the same model-specific authorization effect in this setting. In the revised version, we will further emphasize that the construction is model-agnostic and scales naturally to large vision-language encoders, and we will add a short paragraph and explicit pointer to the extended VLM demo in our GitHub repository to make this generalizability clearer.
>
> | Model                  | Overall |  AR   |  CP   | FP-C  | FP-S  |  LR   |  RR   |
> | ---------------------- | :-----: | :---: | :---: | :---: | :---: | :---: | :---: |
> | InternVL3 (Authorized) |  72.6%  | 77.8% | 82.0% | 58.3% | 78.9% | 50.9% | 67.3% |
> | Qwen2.5-VL             |  18.3%  | 29.2% | 15.6% | 17.0% | 12.8% | 17.3% | 22.3% |
>
>
> ---
> ### View our VLM demo video at (line 501): https://github.com/model-specific/non-transferable-examples

---

### Author Response · Authors · 2025-12-04
**Summary for AC**

We thank the reviewers for their insightful reviews. Below, we briefly summarise the reviews and our post-rebuttal status for the AC.

---
### $\\textcolor{orange}{\\text{Review Highlights}}$:
We are delighted that the reviewers describe our work in the following terms:

**Reviewer** $\\textcolor{orange}{\\text{Qwfv}}$:
> - 'a new protection paradigm'
> - 'tackles a novel and societally relevant problem'
> - 'clear theoretical analysis'

**Reviewer** $\\textcolor{orange}{\\text{94U8}}$:
> - 'novel and interesting'
> - 'address a pressing gap in AI governance'

**Reviewer** $\\textcolor{orange}{\\text{8FC1}}\\textcolor{blue}{\\text{(6 → 8)}}$,
> - 'I really like the idea'
> - 'great observation that worth leaveraging in this problem domain'
> - 'solid theoreticals analysis'
> - 'mathematically sound'
> - **'Great! The author has addressed all my concerns, I will raise my score.'**  ($\\textcolor{orange}{\\text{5 days}}$ *before the incident*)


**Reviewer** $\\textcolor{orange}{\\text{wXeA}}$:
> - 'genuinely novel conceptual contribution'
> - 'simple, efficient, and broadly applicable'
> - 'attractive for real-world deployment'
> - 'solid mathematical analysis'
> - In the discussion, wXeA acknowledged that the main concerns were addressed just before the incident


### $\\textcolor{orange}{\\text{Overall}}$:
|  | |
|-------------------------------------|---------|
| Rebuttal Status | All questions and weaknesses from the reviewers have been perfectly addressed and incorporated into the revision; many points (*e.g.,* reconstruction attacks, VLMs) were in fact **already** in the original version but not clearly noticed and are now more explicitly highlighted. |
| Shared Concern | The only main shared concern ($\textcolor{orange}{\text{Qwfv}}$, $\textcolor{orange}{\text{94U8}}$, $\textcolor{orange}{\text{8FC1}}$) was that computational cost was not well emphasised in the original presentation; ***in fact this is a key strength of the method***, and it is now well emphasised that the method has $\textcolor{orange}{\text{negligible overhead (< 1 μs processing time)}}$. |
| Demo Available | Demo: https://github.com/model-specific/non-transferable-examples |



Paper #10781 Authors

---

### Note · Program_Chairs · 2025-12-08
**Submission Desk Rejected by Program Chairs**

Desk rejected because of the following hallucinated citation:
Dinghuai Zhang, Yang Song, Inderjit Dhillon, and Eric Xing. Defense against adversarial attacks
using spectral regularization. In International Conference on Learning Representations (ICLR),
2020.